# Global, regional, and national burdens of *Clostridioides difficile* infection over recent decades: a trend analysis informed by the Global Burden of Disease Study

Jianmei Zhou,[1,2] Jie Zhu,[1,2] Pengyue Zhang,[1,2] Chunhui Tao,[1,2] Xiaodan Hong,[1] Zhenhua Zhang[1,3]

**ABSTRACT** This study aimed to assess the global burden of *Clostridioides difficile* infection (CDI) from 1990 to 2019, focusing on disability-adjusted life years (DALYs) rates, mortality, and trends. Data were extracted from the Global Burden of Disease Study 2019 and analyzed globally, regionally, and nationally by age, sex, region, and socio-demographic index (SDI). Measures included age-standardized DALYs rate (ASDR), mortality rate (ASMR), and average annual percentage changes (AAPCs). Decomposition analysis and Bayesian age-period-cohort model were used to evaluate factors affecting CDI trends and predict future progress, respectively. Globally, the overall ASDR and ASMR of CDI showed an increasing trend ($AAPC_{ASDR}$ = 1.39, 95% CI: 1.23–1.55; $AAPC_{ASMR}$ = 2.79, 95% CI: 2.66–2.93). High SDI countries showed the highest ASDR (18.86, 95% CI: 17.46–20.24) and ASMR (0.99, 95% CI: 0.87– 1.11), with the fastest growth rate ($AAPC_{ASDR}$ = 2.84, 95% CI: 2.64–3.04; $AAPC_{ASMR}$ = 4.26, 95% CI: 3.98–4.55). Conversely, the low SDI regions exhibit negative growth; however, some low-middle SDI regions, such as South Africa, experienced a heavy disease burden. While most of the disease burden occurs in people over 70 years of age, the burden of children under 5 years of age should also be considered. Moreover, the increased burden on high SDI regions is primarily driven by epidemiological changes. CDI burden has risen globally, particularly in high SDI regions. Moreover, clinicians should take care to consider the burden in individuals under 5 years of age.

**IMPORTANCE** The global burden of *Clostridioides difficile* infection (CDI) is increasing, with notable disparities across regions, age groups, and socioeconomic levels. The higher mortality and disability risks, particularly among older adults, children under 5, and in high socio-demographic index regions, highlight the urgent need for targeted public health interventions and policy adjustments to address these vulnerabilities and reduce the impact of CDI on global health.

**KEYWORDS** *Clostridioides difficile* infection, global burden, DALYs, decomposition analysis, projection analysis

Clostridioides (Clostridium) difficile is an obligate anaerobic, spore-forming, gram-positive bacillus that poses a significant burden on global healthcare by causing healthcare-associated infections and antibiotic-associated diarrhea (1). The primary clinical manifestation of *Clostridioides difficile* infection (CDI) is diarrhea, often accompanied by nausea, vomiting, fever, dehydration, electrolyte disturbances, and acidosis. Some patients may develop more severe colitis, such as pseudomembranous colitis, fulminant colitis, and in extreme cases, complications such as colonic perforation, toxic megacolon, sepsis, and death (2, 3). For CDI treatment, the widespread use of antibacterial drugs has led to a significant decrease in the susceptibility of *C. difficile* to these

**Peer Reviewer** Bobby G. Warren, Duke University, Durham, North Carolina, USA

Address correspondence to Zhenhua Zhang, zzh1974cn@163.com.

Jianmei Zhou and Jie Zhu contributed equally to this article. The author order was determined by type of their contributions.

The authors declare no conflict of interest.

agents, leading to suboptimal treatment outcomes (4–6). Antimicrobial therapy for initial CDI episodes may alter the patient's intestinal microbiota, thereby potentially increasing the risk of recurrence (7, 8). Following CDI onset, 30% of patients experience recurrence, and among those with multiple recurrences, 60% experience further recurrences (9). The recurrent CDI poses a challenge in the field of medicine.

In recent years, the incidence and severity of CDI have been on the rise globally, mostly due to the increasing use of antibiotics, immunosuppressants, corticosteroids, and chemotherapy drugs, as well as the emergence and epidemic spread of highly virulent strains such as RT 027 (10). This trend is particularly noticeable in the developed countries. The most recent data from the Centers for Disease Control and Prevention (CDC) show that there were 223,900 cases of *C. difficile* infection and 12,800 deaths in 2019 (11). Hospital-acquired *C. difficile* infections result in more than a fourfold increase in hospitalization costs, adding approximately $1.5 billion in annual expenditures in the United States alone (12, 13). Thus, it is evident that CDI imposes a significant burden on healthcare systems in terms of both health and financial implications.

Although some research has focused on the burden of CDI in specific regions or countries, there is still a limited understanding of the global burden and epidemiological trends. Due to changes in gut microbiota, immune responses, and susceptibility to underlying conditions, the elderly and children are more susceptible to infection. In the elderly, immunosenescence—characterized by reduced T cell and B cell function, increased inflammatory responses, and diminished response to new antigens—increases their vulnerability to infections (14). The gut microbiota and immune systems of infants and newborns are easily influenced by factors such as mode of delivery, feeding practices, and environmental exposures, making them susceptible to colonization by *C. difficile* (15). Therefore, we need to develop a clear understanding of the burden and trends of CDI in the population. Leveraging the data from the Global Burden and Disease Study (GBD) 2019, we conducted an investigation into mortality rate and disability-adjusted life years (DALYs) rate associated with CDI at the global, regional, and national levels to identify regions or populations with heavy or sustained disease burden. Our study aims to uncover differences in CDI between different regions and populations, inform targeted policies and interventions to alleviate the burden of CDI, and provide deeper insights and scientific support.

## MATERIALS AND METHODS

### Data source

The GBD 2019 collected epidemiological data on 204 countries and territories from 1990 to 2019, providing comprehensive quantitative statistics of 369 diseases and injuries, as well as 87 risk factors. We extracted raw data on CDI from the Global Health Data Exchange (GHDx, http://ghdx.healthdata.org), including estimating DALYs and mortality in terms of absolute number, age-standardized rate (ASR). The CDI data in the GBD study were collected from multiple sources, including scientific literature, discharge records (from multiple countries), hospitalization records, outpatient data, and insurance claims. For *C. difficile*, GBD used the following search string to supplement incidence data on *C. difficile*: "*clostridium difficile*" AND diarrhea[title/abstract] AND (epidemiolog* OR incidence OR prevalence) AND (("2017/06/05"[PDat]: "2019/2/7"[PDat])) NOT (animals [MeSH] NOT humans [MeSH]). They identified 185 studies, of which five met their inclusion criteria. They extracted data points for location, sex, year, and age. These data were processed and analyzed using various modeling methods such as the Cause of Death Ensemble model (CODEm), DisMod-MR, and spatiotemporal Gaussian process regression (ST-GPR) (16). By excluding other disease-related characteristics, covariates, or comorbidities, we obtained data on mortality or disability due to CDI-related diarrhea globally. These data were stratified by geographical location, socioeconomic development level, gender, and different age groups. Based on the social development index (SDI), countries or regions were categorized into five groups: high SDI (> 0.81),

high-middle SDI (0.70–0.81), middle SDI (0.61–0.69), low-middle SDI (0.46–0.60), and low SDI (<0.46). The world was divided into 21 regions geographically. Additionally, the SDI is calculated based on per capita income, educational level, and total fertility rate, serving as a comprehensive indicator of social and demographic development of countries or regions. SDI values for each country and region from 1990 to 2019 were obtained from the http://ghdx.healthdata.org/record/ihme-data/gbd-2019-socio-demographic-index-sdi-1950-2019. The data for this study were sourced from a publicly available database, requiring no ethical approval or informed consent.

## Disease burden description

DALYs consist of the sum of years lost due to premature death (YLLs) and years lived with disability (YLDs). YLLs refer to the number of years of expected life lost due to premature death, calculated by multiplying cause-age-sex-location-year-specific deaths by the standard life expectancy at the age that death occurred (16). YLDs refer to the number of healthy years lost due to poor health or disability, calculated by multiplying the cause-age-sex-location-year-specific prevalence of sequelae by their respective disability weights for each disease and injury. Disability weight for each cause describes the severity of health loss with that specific cause, ranging from 0 (full health) to 1 (death) (16). Due to potential heterogeneity in CDI burden resulting from variations in population age structures, we employed ASR and average annual percentage changes (AAPCs) to quantify the trends in DALYs and mortality rates associated with CDI. AAPC is used to describe the overall trend over a longer period of time. It is calculated by fitting a regression line to the natural logarithm of the rates and provides a summary measure of trend over time. According to different weights to age compositions, we obtained age-standardized DALYs rate (ASDR) and age-standardized mortality rate (ASMR) along with their AAPC, and the 95% confidence intervals (CIs) for AAPC were calculated using a linear regression model. By standardizing to a common age structure, ASDR allows for meaningful comparisons of DALYs between populations with varying age demographics. The ASMR is a mortality rate that has been adjusted to a standard age distribution. This adjustment allows for fair comparisons between populations with different age structures by eliminating the effects of age distribution differences.

## Joint-point analysis

To depict the continuous changes in disease burden, joint-point regression analysis was employed to analyze the time trends of ASDR and ASMR of CDI across different SDI regions (17, 18). This method identifies significant trend changes (known as joint points) and segments the overall trend into multiple sub-segments based on these observed joint points. The annual percentage change (APC) and 95% CI were estimated by log-linear regression to further evaluate the epidemiological trends in each sub-segment.

## Decomposition analysis

We employed the decomposition method invented by Das Gupta to quantify the impact of age structure, population growth, and epidemiological changes on the mortality and DALYs of CDI (19). The decomposition analysis provided valuable insights into the underlying drivers of disease dynamics over time. Age structure refers to the distribution of different age groups within a population. Population growth refers to changes in the total number of people in a region or country. Epidemiological changes refer to shifts in disease incidence, prevalence, and mortality rates, which can be influenced by various factors, including but not limited to pathogen variations, the effectiveness of public health interventions, lifestyle changes, environmental factors, and socioeconomic changes. This methodological approach facilitated a nuanced understanding of the interplay between demographic factors, population dynamics, and epidemiological shifts in shaping the burden of CDI, thereby informing targeted interventions and policies aimed at reducing the disease burden and improving health outcomes.

## Projection analysis

We predicted the disease burden of CDI from 2020 to 2030 based on the Bayesian age-period-cohort (BAPC) model (20, 21). The BAPC model predicts disease DALYs rate and mortality by accounting for the effects of age, time period, and birth cohort, as well as their interactions. Utilizing Bayesian statistical methods, the model incorporates prior information to flexibly model and estimate uncertainty. By integrating observed data on age, time period, and birth cohort, the BAPC model captures how these factors influence disease incidence rates and forecasts future trends.

## Statistical analysis

Variables were expressed in number, percentage, and ratio. Pearson correlation analyses were used to evaluate the associations between ASR and SDI in 2019, AAPC and the mean SDI of 204 countries and territories. We compared the differences in the number of deaths between the <5 age group and other age groups using the paired Mann-Whitney rank-sum test by year. Additionally, to ensure fairness, we conducted a paired Mann-Whitney rank-sum test on the differences in DALYs between adjacent child age groups (<5 age subtraction 5–9 years vs 5–9 years subtraction 10–14 years). All statistical analyses and visualizations of this study were performed in R v.4.1.2 software. $P < 0.05$ was considered statistically significant.

## RESULTS

### Global trends in the distribution of CDI burden in different regions

The global burden of CDI was assessed using various metrics, including DALYs, age-standardized DALYs rate (ASDR), ASMR, AAPCs, and sociodemographic index (SDI).

Globally, the disease burden of CDI has increased over the past three decades. The number of DALYs increased from 428,564 (306,021, 605,835) in 1990 to 870,814 (722,988, 1,052,360) in 2019, and the number of deaths increased from 8,321 (6,637, 10,469) in 1990 to 32,134 (28,131, 36,549) in 2019. The overall ASDR and ASMR of CDI showed an increasing trend ($AAPC_{ASDR}$ = 1.39, 95% CI: 1.23–1.55; $AAPC_{ASMR}$ = 2.79, 95% CI: 2.66–2.93, Table 1).

Regionally, from 1990 to 2019, ASDR and ASMR increased in most of the 21 regions, with the most significant increase observed in Southern Latin America ($AAPC_{ASDR}$ = 8.51, 95% CI: 8.03–9.00; $AAPC_{ASMR}$ = 11.08, 95% CI: 8.03–9.00, Table 1; Fig. 1A and B; File S1). However, ASDR and ASMR decreased in four regions, with Oceania showing the most pronounced decline ($AAPC_{ASDR}$ = −3.99, 95% CI: −4.17 to −3.81; $AAPC_{ASMR}$ = −3.84, 95% CI: −3.97 to −3.71, Table 1 ; Fig. 1C and D; File S1), followed by Western Sub-Saharan Africa.

Nationally, most developed countries have a greater CDI burden than developing countries, but there are exceptions, such as South Africa and Venezuela, where the CDI burden was also significant. In 2019, countries with high ASDR and ASMR were mainly concentrated in Southern Sub-Saharan Africa, North America, and Western Europe. For example, South Africa had the highest ASDR, and the United States had the highest ASMR (South Africa$_{ASDR}$ = 75.12, 95% CI: 56.74–97.82; the United States$_{ASMR}$ = 1.72, 95% CI: 1.54–1.87, Fig. 1A and B). Over the past 30 years, the vast majority of countries have experienced increases in both ASDR and ASMR. However, a few countries and regions where ASMR and ASDR have shown a significant downward trend are mainly concentrated in Western Sub-Saharan Africa and Oceania. For example, Zimbabwe ($AAPC_{ASMR}$ = −5.89, 95% CI: −7.97 to −3.77; $AAPC_{ASDR}$ = −5.87 95% CI: −3.40 to −8.29, Fig. 1C and D). Interestingly, when observing from a map perspective, only Brazil exhibited a downward trend throughout the Americas.

### The distribution characteristics of CDI burden across different SDI regions

Among the five GBD SDI quintiles in 2019, high SDI countries showed the highest ASDR and ASMR (ASDR = 18.86, 95% CI: 17.46–20.24; ASMR = 0.99, 95% CI: 0.87–1.11, Table 1),

**TABLE 1** Global mortality and death of CDI and their AAPC by gender, SDI level, and region[a]

| Characteristics | DALY 1990 | | DALY 2019 | | | Mortality 1990 | | Mortality 2019 | | |
|---|---|---|---|---|---|---|---|---|---|---|
| | No (95% UI) | ASDR (/10,000) (95% UI) | No. (95% UI) | ASDR (/10,000) (95% UI) | AAPC, % (95% CI) | No. (95% UI) | ASMR (/10,000) (95% UI) | No. (95% UI) | ASMR (/10,000) (95% UI) | AAPC, % (95% CI) |
| Global | 428,564 (306,021–605,835) | 7.79 (5.78–10.66) | 870,814 (722,988–1052,360) | 11.69 (9.62–14.25) | 1.39 (1.23–1.55) | 8,321 (6,637–10,469) | 0.19 (0.16–0.23) | 32,134 (28,131–36,549) | 0.43 (0.37–0.49) | 2.79 (2.66–2.93) |
| Gender | | | | | | | | | | |
| Male | 233,560 (165,584–330,198) | 8.43 (6.27–11.55) | 446,221 (367,322–546,687) | 12.36 (10.18–15.20) | 1.31 (1.09–1.54) | 4,194 (3,263–5,406) | 0.45 (0.39–0.52) | 14,800 (12,892–16,810) | 0.45 (0.39–0.52) | 2.81 (2.66–2.96) |
| Female | 195,004 (139,604–274,349) | 7.11 (5.26–9.81) | 424,594 (350,860–516,161) | 11.06 (8.96–13.73) | 1.50 (1.33–1.67) | 4,127 (3,371–5,113) | 0.41 (0.35–0.48) | 17,334 (14,873–19,960) | 0.41 (0.35–0.48) | 2.83 (2.67–2.98) |
| SDI rank | | | | | | | | | | |
| High SDI | 68,757 (62,426–75,462) | 8.29 (7.40–9.26) | 314,086 (284,804–341,436) | 18.86 (17.46–20.24) | 2.84 (2.64–3.04) | 2,914 (2,586–3,249) | 0.30 (0.27–0.33) | 20,608 (17,624–23,649) | 0.99 (0.87–1.11) | 4.26 (3.98–4.55) |
| High-middle SDI | 101,834 (72,525–140,019) | 9.25 (6.63–12.79) | 142,585 (118,255–173,282) | 11.27 (9.10–14.15) | 0.58 (0.41–0.75) | 1,781 (1,361–2,271) | 0.17 (0.13–0.22) | 4,842 (4,101–5,665) | 0.30 (0.25–0.35) | 1.88 (1.64–2.12) |
| Middle SDI | 176,402 (117,864–260,705) | 9.10 (6.17–13.24) | 293,817 (215,655–389,036) | 13.83 (10.08–18.43) | 1.41 (1.13–1.68) | 2,521 (1,737–3,553) | 0.14 (0.10–0.19) | 4,921 (3,722–6,306) | 0.22 (0.17–0.28) | 1.45 (1.33–1.57) |
| Low-middle SDI | 57,193 (34,926–95,147) | 3.97 (2.43–6.38) | 89,736 (56,488–140,282) | 4.97 (3.13–7.70) | 0.75 (0.34–1.15) | 775 (477–1,247) | 0.06 (0.04–0.09) | 1,326 (847–1,956) | 0.08 (0.05–0.11) | 0.88 (0.56–1.21) |
| Low SDI | 24,151 (14,072–42,349) | 3.36 (2.03–5.56) | 30,206 (15,114–57,011) | 2.13 (1.11–3.83) | −1.59 (−1.88 to −1.31) | 326 (195–548) | 0.05 (0.03–0.08) | 424 (221–766) | 0.03 (0.02–0.06) | −1.38 (−1.64 to −1.11) |
| GBD regions | | | | | | | | | | |
| Africa | | | | | | | | | | |
| Central Sub-Saharan Africa | 2,299 (1,089–4,368) | 3.04 (1.57–5.36) | 5,147 (2,763–9,107) | 3.03 (1.69–5.18) | −0.11 (−0.36 to 0.14) | 31 (16–56) | 0.05 (0.03–0.08) | 71 (39–122) | 0.05 (0.03–0.08) | −0.13 (−0.34 to 0.09) |
| Eastern Sub-Saharan Africa | 9,392 (5,098–17,142) | 3.52 (1.97–6.03) | 20,435 (11,515–36,345) | 3.90 (2.29–6.58) | 0.21 (0.00–0.42) | 125 (68–219) | 0.05 (0.03–0.09) | 283 (164–486) | 0.06 (0.04–0.10) | 0.37 (0.19–0.55) |
| Southern Sub-Saharan Africa | 31,060 | 51.08 (37.56–67.15) | 44,089 | 53.31 (40.08–69.37) | 0.22 (−0.14 to 0.59) | 457 (339–595) | 0.83 (0.63–1.06) | 703 (531–902) | 0.88 (0.67–1.12) | 0.22 (−0.19 to 0.64) |

TABLE 1 Global mortality and death of CDI and their AAPC by gender, SDI level, and region[a] (Continued)

| Characteristics | DALY 1990 No (95% UI) | DALY 1990 ASDR (/10,000) (95% UI) | DALY 2019 No. (95% UI) | DALY 2019 ASDR (/10,000) (95% UI) | DALY 2019 AAPC, % (95% CI) | Mortality 1990 No. (95% UI) | Mortality 1990 ASMR (/10,000) (95% UI) | Mortality 2019 No. (95% UI) | Mortality 2019 ASMR (/10,000) (95% UI) | Mortality 2019 AAPC, % (95% CI) |
|---|---|---|---|---|---|---|---|---|---|---|
| Western Sub-Saharan | (22,656–41,886) | | (33,154–57,579) | | | | | | | |
| Africa | 53,289 (37,515–75,666) | 21.61 (15.63–29.52) | 66,688 (44,511–98,419) | 11.41 (7.67–16.33) | −2.55 (−2.87 to −2.23) | 750 (540–1,036) | 0.35 (0.26–0.48) | 922 (620–1,320) | 0.18 (0.12–0.24) | −2.69 (−2.98 to −2.40) |
| North Africa and Middle East | 7,174 (3,525–13,484) | 1.61 (0.82–2.91) | 23,689 (15,731–34,878) | 3.88 (2.60–5.65) | 3.03 (2.47–3.60) | 98 (50–177) | 0.02 (0.01–0.04) | 395 (272–548) | 0.07 (0.05–0.10) | 3.7 (3.23–4.17) |
| Asia | | | | | | | | | | |
| Central Asia | 1,750 (895–3,225) | 2.23 (1.17–3.94) | 2,976 (1,815–4,617) | 3.15 (1.93–4.84) | 1.27 (0.60–1.95) | 26 (14–45) | 0.04 (0.02–0.06) | 47 (30–69) | 0.05 (0.03–0.08) | 1.42 (0.81–2.03) |
| East Asia | 57,930 (36,951–88,559) | 4.76 (3.07–7.29) | 78,315 (58,995–107,386) | 6.50 (4.95–8.65) | 1.07 (0.67–1.47) | 906 (605–1,293) | 0.08 (0.05–0.11) | 1,927 (1,442–2,633) | 0.13 (0.10–0.17) | 1.68 (1.30–2.07) |
| South Asia | 60,446 (34,984–102,114) | 4.46 (2.63–7.19) | 80,483 (47,649–128,451) | 4.40 (2.64–7.02) | −0.06 (−0.53 to 0.42) | 846 (501–1,364) | 0.07 (0.04–0.10) | 1,311 (808–1,978) | 0.08 (0.05–0.11) | 0.27 (−0.15 to 0.69) |
| Southeast Asia | 71,988 (49,106–105,337) | 13.4 (9.28–18.88) | 79,816 (55,507–110,406) | 12.45 (8.71–17.41) | −0.28 (−0.60 to 0.03) | 1,053 (732–1,459) | 0.22 (0.15–0.29) | 1,387 (983–1,859) | 0.22 (0.16–0.29) | −0.03 (−0.23 to 0.17) |
| High-income Asia Pacific | 18,041 (15,702–20,569) | 11.38 (9.82–13.14) | 40,114 (33,085–47,615) | 12.95 (11.21–15.09) | 0.41 (0.13–0.70) | 634 (518–762) | 0.37 (0.30–0.44) | 2,650 (2,066–3,277) | 0.54 (0.44–0.64) | 1.30 (0.89–1.71) |
| Europe | | | | | | | | | | |
| Central Europe | 2,841 (2,137–3,705) | 2.56 (1.87–3.38) | 5,887 (4,706–7,267) | 5.08 (4.18–6.09) | 2.31 (1.86–2.77) | 59 (44–76) | 0.05 (0.04–0.06) | 202 (148–268) | 0.12 (0.10–0.15) | 3.35 (3.11–3.6) |
| Eastern Europe | 7,595 (4,755–10,993) | 3.75 (2.35–5.53) | 8,166 (6,360–10,041) | 4.35 (3.41–5.38) | 0.65 (−0.50 to 1.82) | 129 (81–184) | 0.06 (0.04–0.09) | 211 (159–268) | 0.09 (0.07–0.11) | 1.51 (0.44–2.59) |
| Western Europe | 31,426 (28,637–34,133) | 6.96 (6.22–7.81) | 112,471 (98,984–126,644) | 13.72 (12.51–14.98) | 2.29 (1.98–2.60) | 1,700 (1,474–1,919) | 0.3 (0.27–0.34) | 8,389 (6,900–9,955) | 0.78 (0.66–0.90) | 3.24 (2.95–3.52) |
| Latin America and the Caribbean | | | | | | | | | | |
| Andean Latin America | 767 (341–1,497) | 1.65 (0.77–3.11) | 10,148 (6,459–15,420) | 15.99 (10.18–24.21) | 3.96 (3.54–4.38) | 11 (5–20) | 0.03 (0.01–0.05) | 160 (109–227) | 0.26 (0.18–0.37) | 7.82 (6.72–8.93) |

**TABLE 1** Global mortality and death of CDI and their AAPC by gender, SDI level, and region[a] (Continued)

| Characteristics | DALY 1990 | | DALY 2019 | | | Mortality 1990 | | Mortality 2019 | | |
|---|---|---|---|---|---|---|---|---|---|---|
| | No (95% UI) | ASDR (/10,000) (95% UI) | No. (95% UI) | ASDR (/10,000) (95% UI) | AAPC, % (95% CI) | No. (95% UI) | ASMR (/10,000) (95% UI) | No. (95% UI) | ASMR (/10,000) (95% UI) | AAPC, % (95% CI) |
| Caribbean | 1,659 (1,167–2,286) | 4.50 (3.23–6.01) | 3,527 (2,717–4,589) | 7.71 (5.92–10.1) | 1.79 (1.18–2.41) | 29 (21–37) | 0.09 (0.07–0.11) | 86 (67–109) | 0.18 (0.14–0.22) | 2.5 (1.9–3.1) |
| Central Latin America | 11,865 (6,762–20,211) | 5.73 (3.42–9.54) | 66,337 (47,392–88,772) | 29.19 (20.78–39.45) | 5.71 (5.09–6.34) | 160 (96–263) | 0.09 (0.05–0.13) | 933 (681–1,239) | 0.41 (0.29–0.54) | 5.52 (4.82–6.21) |
| Southern Latin America | 762 (401–1,344) | 1.51 (0.81–2.65) | 11,413 (9,226–13,805) | 15.65 (12.83–18.88) | 8.51 (8.03–9.00) | 14 (8–22) | 0.03 (0.02–0.05) | 477 (356–621) | 0.59 (0.44–0.75) | 11.08 (10.38–11.79) |
| Tropical Latin America | 39,949 (26,594–57,431) | 22.38 (15.09–31.92) | 31,470 (22,545–42,809) | 16.48 (11.73–22.85) | −1.02 (−1.48 to −0.56) | 534 (359–751) | 0.31 (0.21–0.44) | 556 (404–778) | 0.27 (0.20–0.37) | −0.43 (−0.69 to −0.17) |
| Oceania and Australasia | | | | | | | | | | |
| Oceania | 630 (443–917) | 8.39 (6.01–11.57) | 379 (219–645) | 2.59 (1.54–4.16) | −3.99 (−4.17 to −3.81) | 9 (7–13) | 0.15 (0.11–0.19) | 6 (4–9) | 0.05 (0.03–0.07) | −3.84 (−3.97 to −3.71) |
| Australasia | 926 (831–1,026) | 4.55 (4.03–5.12) | 3,585 (3,176–4,043) | 8.76 (7.79–9.85) | 2.35 (2.02–2.69) | 43 (38–48) | 0.20 (0.18–0.23) | 243 (203–284) | 0.46 (0.39–0.52) | 2.95 (2.58–3.33) |
| North America | | | | | | | | | | |
| High-income North America | 16,773 (14,744–18,550) | 5.9 (5.02–6.67) | 175,679 (162,342–187,667) | 30.08 (28.12–31.90) | 5.83 (5.49–6.18) | 707 (643–749) | 0.21 (0.19–0.22) | 11,175 (9,833–12,402) | 1.68 (1.51–1.84) | 7.71 (7.11–8.30) |

[a]AAPC, average annual percentage change; ASDR, age-standardized disability-adjusted life year; ASMR, age-standardized mortality rate; CDI, *Clostridioides difficile* infection; DALYs disability-adjusted life years; SDI, sociodemographic index.

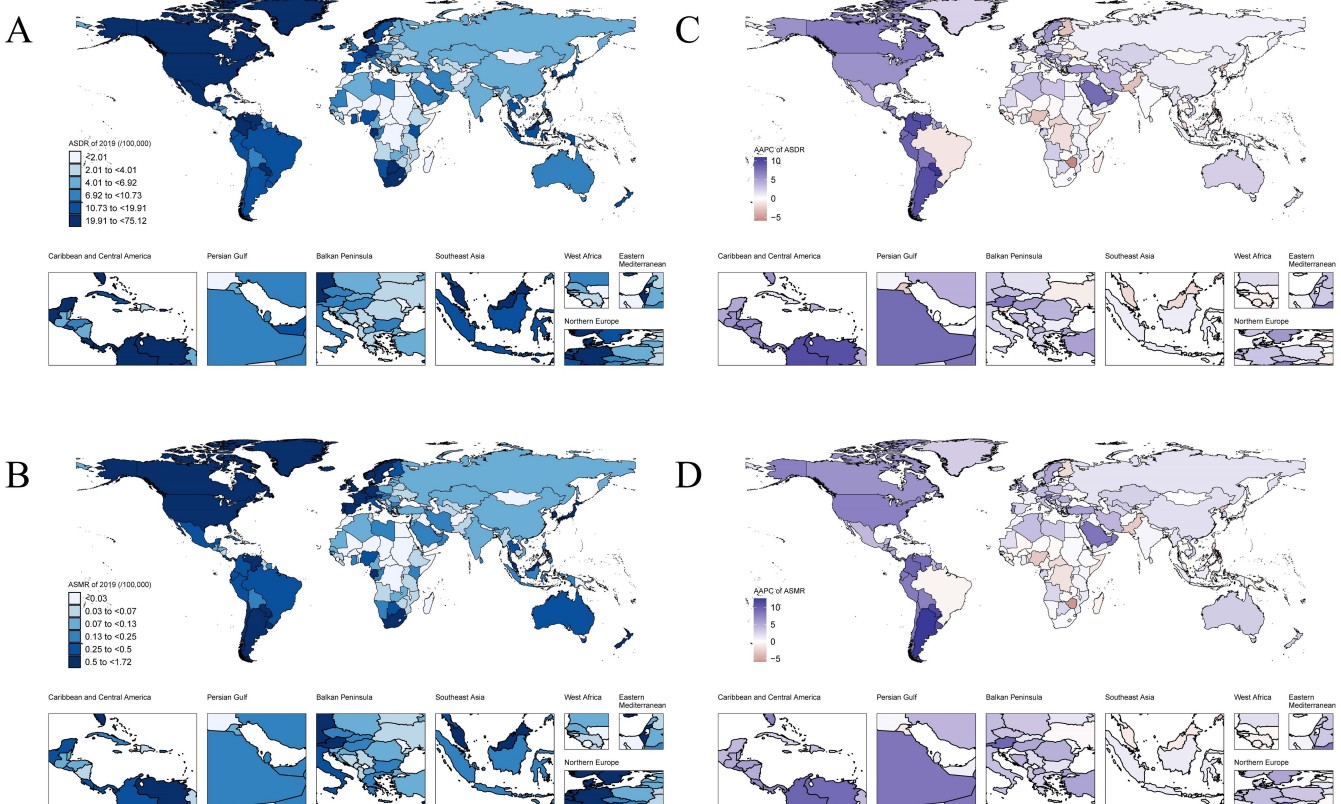

**FIG 1** The distribution of ASR from CDI in 204 countries and territories. (A) The distribution of ASR from CDI in 204 countries and territories Regional distribution of the ASDR from CDI in 2019. (B) Regional distribution of the ASMR from CDI in 2019. (C) Regional distribution of trends in ASDR from 1990 to 2019. (D) Regional distribution of trends in ASMR from 1990 to 2019. ASDR, age-standardized disability-adjusted life year; ASMR, age-standardized mortality rate; CDI, *Clostridioides difficile* infection. Maps created using the R package "easyGBDR" to process and visualize data from the GBD database.

with the fastest growth rate (AAPC$_{ASDR}$ = 2.84, 95% CI: 2.64–3.04; AAPC$_{ASMR}$ = 4.26, 95% CI: 3.98–4.55). Conversely, the low SDI have the lightest disease burden, exhibiting negative growth (AAPC$_{ASDR}$ = −1.59, 95% CI: −1.88 to −1.31; AAPC$_{ASMR}$ = −1.38, 95% CI: −1.64 to −1.11, Table 1). We found that the ASDR, ASMR, and their AAPC were significantly positively correlated with SDI levels in 2019 ($R$ = 0.535, $P < 10^{-4}$, $R$ = 0.701, $P < 10^{-4}$, Fig. 2A and B), and AAPC from 1990 to 2019 ($R$ = 0.171, $P$ = 0.015, $R$ = 0.285, $P < 10^{-4}$, Fig. 2C and D). Then, we investigate the correlation between ASDR, ASMR, and their AAPC in 21 regions (Fig. S1A and B) and 204 countries (Fig. S1C and D). Joint-point analysis indicated that both ASDR and ASMR increased most significantly in High SDI areas, with a sharp rise primarily occurring during the decade from 1997 to 2007. From 2007 to 2016, it plateaued, and since 2016, it has shown a downward trend (Fig. 3A and B; Table S2).

## The distribution characteristics of CDI burden across different age groups

Age groups were divided into the following categories: under 14 years of age, 15–44 years of age, 45–69 years of age, and 70 years of age or older. CDI posed the heaviest disease burden on individuals aged 70 and above, with DALY rates and mortality significantly higher than other age groups (Fig. 4A and B). From 1990 to 2019, globally, ASMR and ASDR increased across all age groups, but the growth rate among those aged 70 and above was significantly higher than other age groups (AAPC$_{ASDR}$ = 4.23, 95% CI: 3.92–4.54; AAPC$_{ASMR}$ = 4.18, 95% CI: 3.98–4.38, Table S1). Figure 3C and D illustrated the changing trends over time among elderly individuals aged 70 and above in five SDI regions. It is evident that in high SDI regions, individuals aged 70 and above had the highest disease burden.

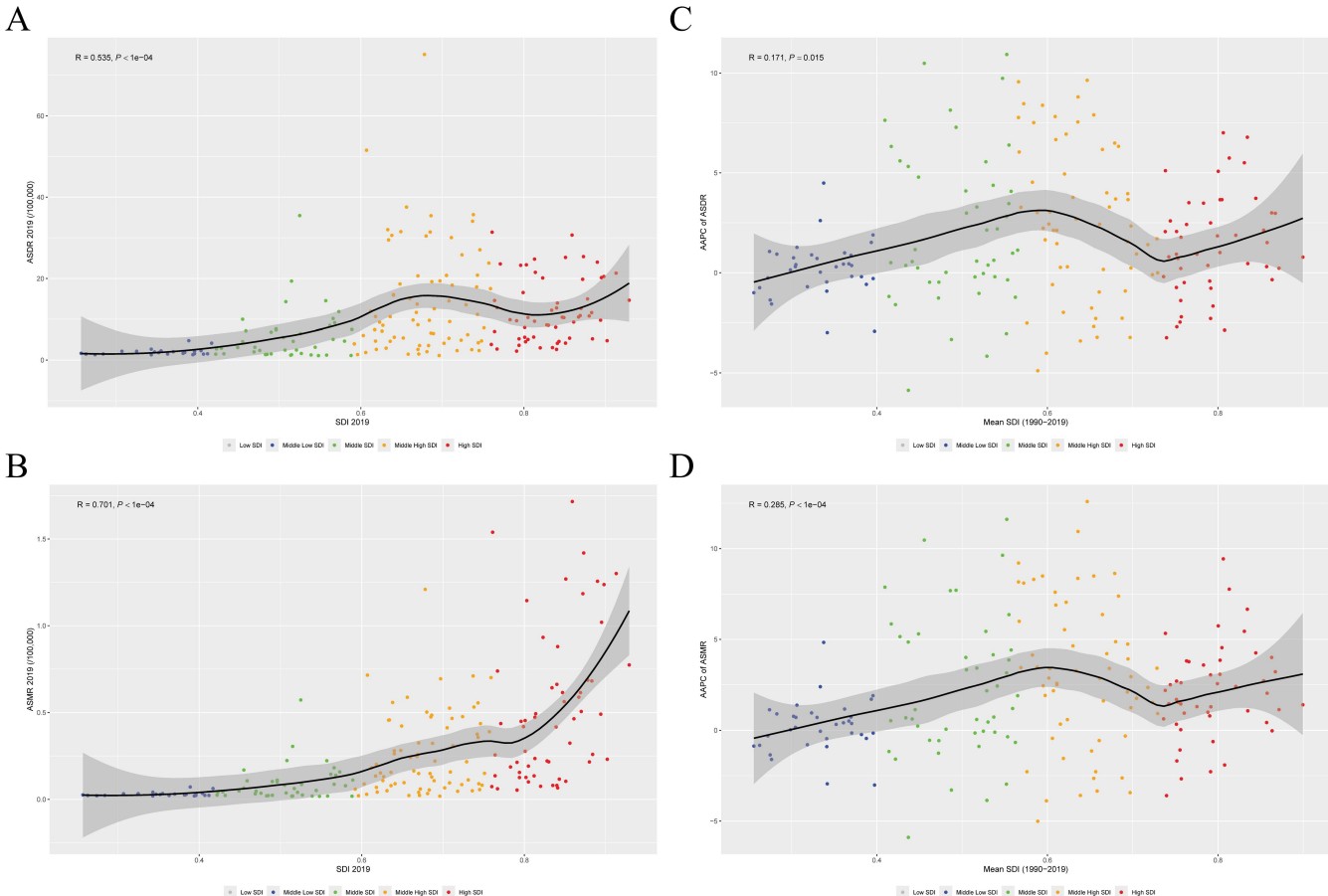

**FIG 2** Association between SDI and ASDR (A), ASMR (B) of CDI in 2019 and corresponding AAPC (C, D) from 1990 to 2019. ASDR, age-standardized disability-adjusted life year; ASMR, age-standardized mortality rate; CDI, *Clostridioides difficile* infection; SDI, sociodemographic index.

When examining SDI levels, ASDR and ASMR showed a declining trend across all age groups in low SDI areas, except for those aged 70 and above. Conversely, in high SDI regions, ASDR and ASMR exhibited an upward trend across all age groups, except for children under 14. Across all SDI levels, individuals aged 70 and above had the largest AAPC. Additionally, considering diarrhea being still the second most common cause of death in children under 5 years old (22, 23), we paid attention to children under 5 years old and found that although their DALYs rates and mortality were relatively low, this age group had the highest number of deaths and DALYs within this vulnerable group of children (<14 years) (*P* < 0.0001). Additionally, the number of deaths in children under 5 years old exceeded that of all other age groups under 45 years old (*P* < 0.0001). (Fig. 4C and D). Over the past 30 years, the ASDR and ASMR for CDI in children under 5 years old showed a global increasing trend, a downward trend in high SDI and low SDI regions, and a rising trend in high-middle SDI, middle SDI, and low-middle SDI regions (Table S1). Figure 3E and F depicted the evolving trends over time among children under 5 years old across five SDI regions. It is apparent that in middle SDI regions, the disease burden among this age group is the most pronounced.

## Decomposition analysis

Our decomposition analysis provided insights into the relative contributions of aging, population growth, and epidemiological changes in DALYs and mortality of CDI according to five SDI regions and 21 GBD regions. High SDI regions had an increase in DALYs and mortality, primarily driven by epidemiological changes, followed by aging (Fig. 5A and C). Two regions with increasing epidemiological change, high-income North America

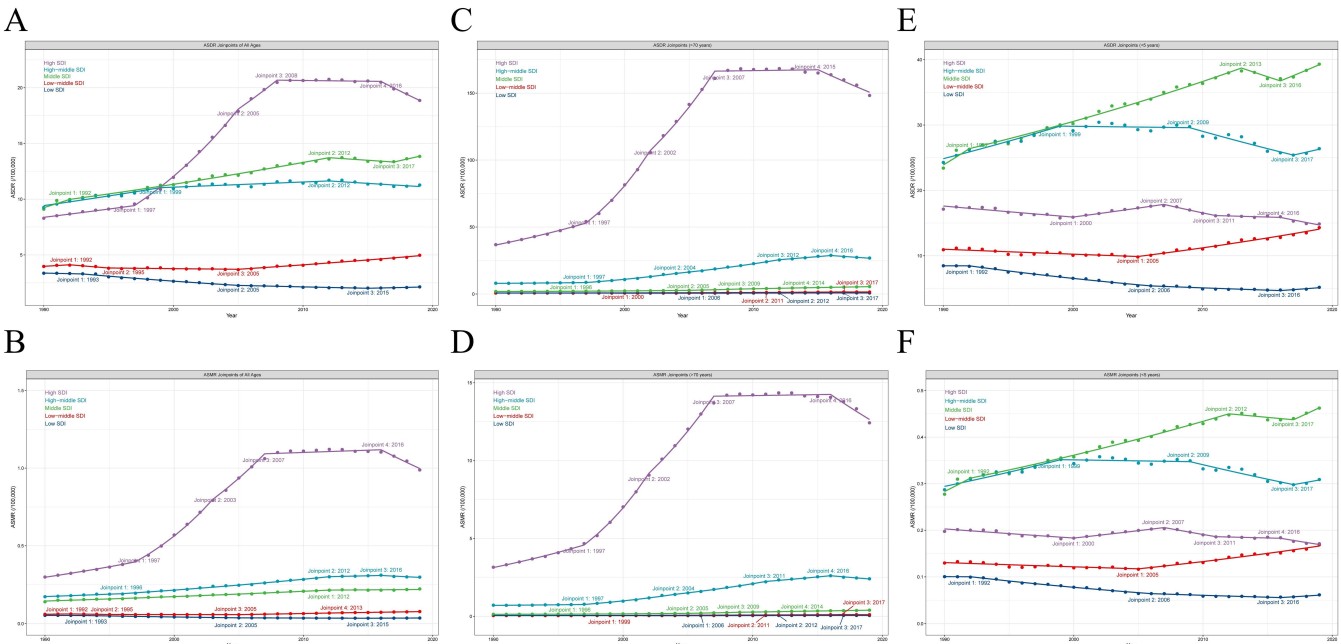

**FIG 3** The changing trends of ASR in five SDI regions of the total population, elderly over 70 years old, and children under 5 years old, from 1990 to 2019. The changing trends of ASDR (A), ASMR (B) in five SDI regions of the total population. The changing trends of ASDR (C), ASMR (D) in five SDI regions of the elderly over 70 years old. The changing trends of ASDR (E), ASMR (F) in five SDI regions of the children under 5 years old. ASDR, age-standardized disability-adjusted life year; ASMR, age-standardized mortality rate; ASR, age-standardized rate; SDI, sociodemographic index.

and Western Europe, had an increase in total CDI DALYs and mortality between 1990 and 2019 (Fig. 5B and D). Regarding gender, there was no significant difference between males and females.

## Projection analysis

Given the heavier burden of CDI in high SDI regions, we focused our projections on this area. From 2020 to 2030, it is anticipated that DALYs and death cases of CDI in children under 5 years old will decrease, as well as ASDR and ASMR (Fig. S2A). However, DALY cases and ASDR in individuals aged over 70 are predicted to increase, while death cases and ASMR are projected to decrease (Fig. S2B and C).

## DISCUSSION

This study investigated the global patterns and trends of CDI burden across different regions and populations over the past 30 years. We found that the overall trend indicated a heavier disease burden in regions with higher SDI, and the burden increasing at a faster rate. Conversely, regions with lower SDI exhibited a decreasing burden. However, some low-middle SDI regions, such as South Africa, experienced a heavy disease burden. Across all age groups, individuals aged 70 and above bore the heaviest burden, and the burden was increasing most rapidly in this age group. Moreover, the number of CDI-related deaths in children under five years old is relatively high. Therefore, it is crucial to increase attention to this issue and implement targeted prevention and treatment measures.

   *C. difficile* was first identified as the pathogen associated with antibiotic-related diarrhea in the late 1970s (24). In the last decade of the 20th century, the incidence of CDI began to rise, becoming a well-known cause of hospital-acquired infections in developed countries. In 2002, the epidemic strain of *C. difficile*, BI/NAP1/027 clone , was first discovered, leading to outbreaks of CDI in more than 20 countries in North America and Europe (25). Subsequently, infections caused by *C. difficile* were reported in multiple

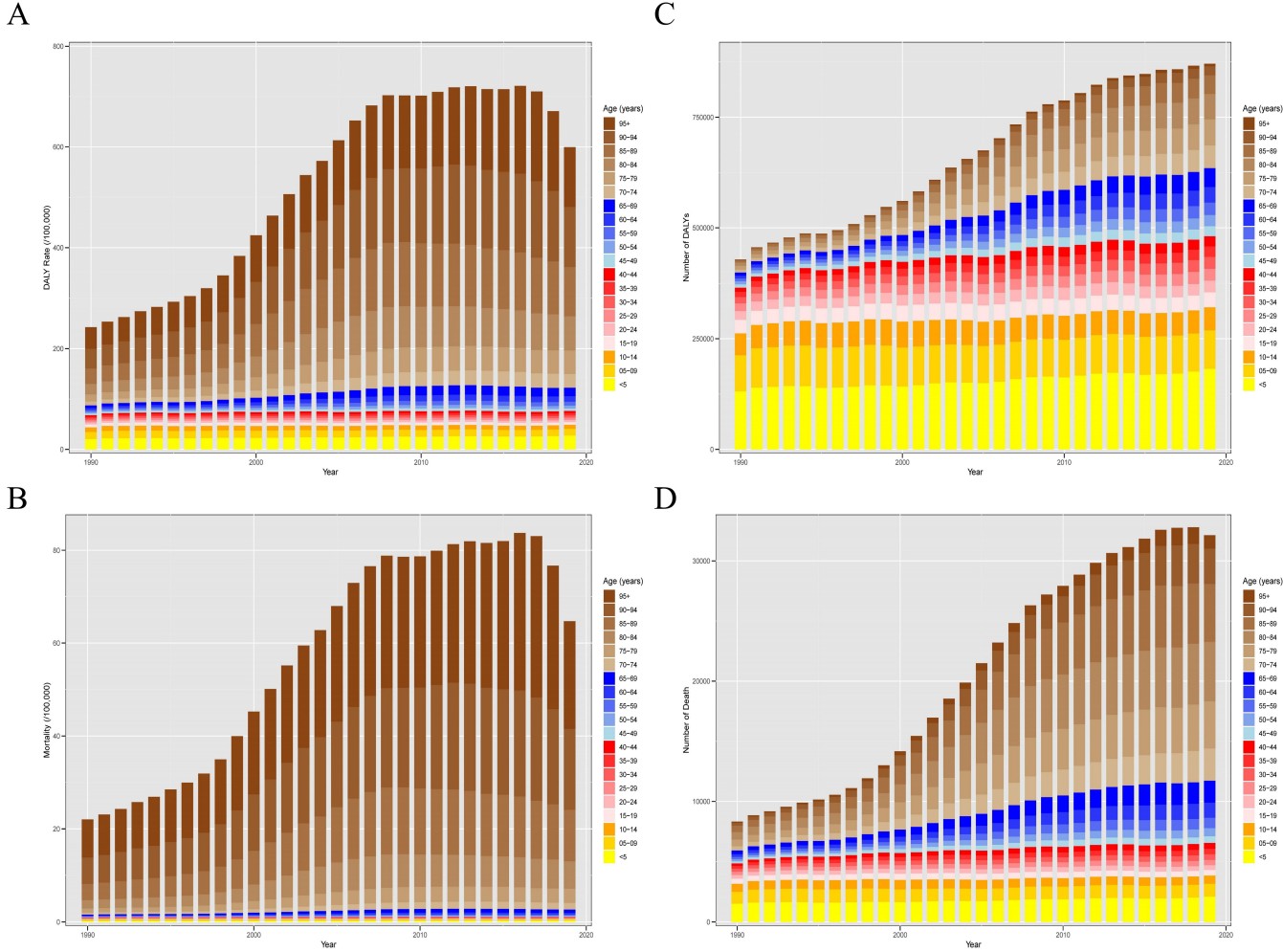

**FIG 4** The rate and cases of different age groups from 1990 to 2019. The DALYs rate (A), mortality (B), and the number of DALYs (C) and death (D) of different age groups from 1990 to 2019. DALYs, disability-adjusted life years.

countries (26, 27). A meta-analysis included healthcare facility-associated CDI incidence data from 41 countries worldwide. The comparison revealed that CDI incidence was higher in high-income North America, which is consistent with our conclusion that high SDI regions tend to bear a heavier burden of CDI (28). Our decomposition analysis revealed that the primary determinant of CDI burden was the epidemiological change of the high SDI regions, rather than aging and population growth. These epidemiological changes may be due to the prevalence of highly virulent strains and antibiotic exposure. The reports of highly virulent strains are mainly concentrated in high SDI regions (29). In addition, high SDI countries have higher antibiotic consumption rates, particularly with a greater frequency of use of broad-spectrum antibiotics such as carbapenems and polymyxins (30).

However, little is known about the impact CDI may have in countries with low or middle SDI. Our study suggested that low SDI regions showed lower CDI burden, which we hypothesize may be related to the use of non-top of the line antibiotics and the greater diversity of the gut microbiota. To our knowledge, antibiotic treatment is the most important risk factor for the development of CDI (31, 32). Studies have shown that antibiotics such as clindamycin, carbapenems, and third- and fourth-generation cephalosporins are most strongly associated with healthcare-associated CDI. However, antibiotics, such as tetracyclines, sulfonamides, and macrolides, have a weaker association with the occurrence of CDI (33). In low SDI countries, more than 60% of antibiotics

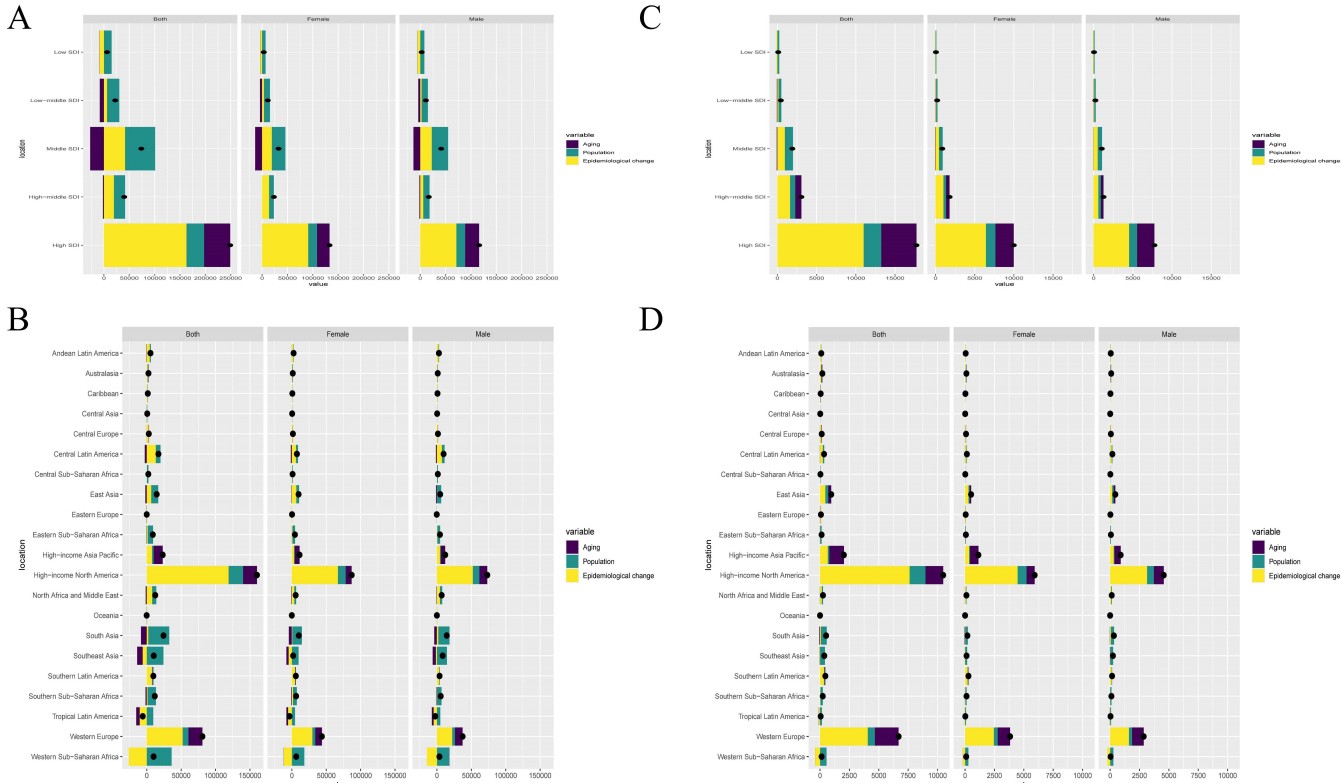

**FIG 5** The decomposition analysis of CDI change in DALYs and mortality by SDI and 21 GBD regions, 1990 to 2019. The decomposition analysis of CDI change in DALYs by SDI (A), 21 regions (B). The decomposition analysis of CDI change in mortality by SDI (C), 21 regions (D). CDI, *Clostridioides difficile* infection; DALYs, disability-adjusted life years; GBD, global burden of disease; SDI, sociodemographic index.

can be purchased directly without a prescription and not necessarily through a pharmacy in many low-middle income countries (34). Antibiotics commonly available in low-SDI regions are often non-top of the line antibiotics, such as tetracyclines and sulfonamides, which are less likely to induce severe CDI. Additionally, the unique composition and diversity of the gut microbiota may be reasons for the lower CDI burden in low SDI regions. The study by De Filippo et al. 35 compared the gut microbiota of children from Europe and rural Africa using high-throughput 16S rDNA sequencing and biochemical analysis. The study found that the higher fiber content in the diet of African children led to significant enrichment of Bacteroides and depletion of Bacillota, with a unique abundance of bacteria from the genus Prevotella. According to some research (36–38), particular species/OTUs of the Bacteroides and Prevotella genera are considered protective, while specific Bacillota species/OTUs, such as Clostridium XI (OTU39), Erysipelotrichaceae (OTU22), and Streptococcus (OTU10), are associated with CDI susceptibility (36). In addition, some animal studies have confirmed that dietary fibers help in resisting CDI (39, 40). Therefore, the higher fiber diet in low SDI regions may promote the growth of beneficial bacteria and increase the diversity of the gut microbiota, thereby enhancing the gut's resistance to CDI.

Regarding the prevention measures and strategies for CD, in high SDI countries, it is necessary to control patient density, promptly detect high-risk pathogens, and strengthen isolation, disinfection, and hand hygiene measures. In addition, it is crucial to continue improving antibiotic prescription practices. For low SDI regions, it is necessary to develop and provide cost-effective *C. difficile* diagnostic tests, formulate reasonable antibiotic management plans, and minimize unnecessary and over-the-counter antibiotic use.

Our study showed that individuals aged 70 and above bore the heaviest CDI burden, which was consistent with findings from previous research (41, 42). Studies have shown

that the high prevalence of CDI is closely related to dysbiosis of the gut microbiota (43, 44). Previous research has shown that gut microbiota imbalance is often caused by factors such as nutritional deficiencies following acute or chronic diseases, long-term and extensive use of antibiotics, and abdominal surgeries (45, 46), conditions that are more common among the elderly. Furthermore, older adults tend to have lower abundance of protective bacterial groups in their gut microbiota in healthcare settings, further increasing their risk of infection (47). In addition, another factor that makes the elderly more susceptible to CDI may be related to impaired gut barrier function (48, 49). Studies have also shown that hospitalized patients who have high levels of anti-toxin A IgG following CDI are less likely to develop diarrhea (50), and serum IgG and IgM responses to toxin A are crucial for protecting patients from CDI and preventing recurrence (51). However, due to the compromised immune system in the elderly, they produce fewer specific IgG antibodies against *C. difficile* toxin A (52, 53), which makes them more susceptible to infection. In addition to the elderly, children under 5 years old are also an age group worthy of attention. Research indicates that the rate of CDI in children aged 1–5 years seems to be increasing (54, 55). It is noteworthy that although the colonization rate of *C. difficile* in infants is high, clinical infections are rare. The resistance of newborns to CDI may be related to the absence of toxin receptors in their immature intestinal mucosa, the lack of downstream signaling pathways, and certain protective factors found in breast milk and the host gut microbiome (56, 57). Additionally, a review study (58) indicates that antibiotic exposure, previous hospitalization, and the presence of underlying comorbid conditions such as inflammatory bowel disease (IBD) are risk factors for CDI in children. We need to focus particularly on these populations. For children under 5 years and elderly individuals over 70 years population in high SDI regions, our predictive study suggests that the disease burden of CDI will reduce over the next 10 years due to the implementation of preventive and therapeutic measures. However, it is worth noting that the number of DALYs cases and ASDR in individuals over 70 years old are still increasing, so efforts should be intensified to manage infections in this population and improve their quality of life.

Our research holds several advantages. By focusing on CDI, we contribute valuable insights into the challenges posed by this infection within healthcare settings. Understanding the epidemiology and burden of CDI is pivotal for guiding healthcare policies and interventions. However, there are several limitations that need to be acknowledged. First, in areas with limited healthcare resources, there may be insufficient detection capacity. Therefore, the reported cases of CDI to public health authorities may be lower than the actual number of cases. Second, previous literature (59) has criticized the DALY metric primarily due to the social and ethical issues associated with age weighting and the allocation of disability weights (DWs), as well as the societal resource allocation and discrimination issues caused by time preferences and future life discounting. GBD2019 eliminated age weighting, adopting equal weighting for all age groups to evaluate disease burden. However, the death or disability of younger populations has a greater impact on social productivity. Our study indirectly demonstrates the objective loss of social productivity caused by CDI by stratifying the data by age group. Moreover, the lack of CDI incidence and prevalence estimates in our study limits our ability to draw precise and comprehensive conclusions about the global burden of CDI. Additionally, the GBD data does not differentiate between the severity of CDI cases, nor can it distinctly separate the burden of hospital-acquired infections from community-acquired infections. Finally, we could not explore the causal relationship between antibiotic use and the burden of CDI in detail, as the GBD data does not include specific prescription information. Therefore, future research should integrate multiple data sources and more detailed analyses to better understand the global burden and influencing factors of CDI.

## Conclusions

We analyzed the global burden and trends of CDI in the past three decades in different regions and countries. The disease burden was heavy in high SDI regions and also

positively correlated with the SDI level. Children under 5 years old and adults over 70 years old are high-risk populations for CDI, especially among those aged 70 and older in relatively high SDI regions. We should focus more on these regions and populations and formulate targeted public health strategies. Meanwhile, in less developed areas, there should be increased testing for high-risk populations and broader epidemiological studies conducted.

## ACKNOWLEDGMENTS

The authors thank the Institute for Health Metrics and Evaluation for sharing valuable GBD data.

This work was supported by Research Fund of Anhui Institute of Translational Medicine (2023zhyx-B10), Outstanding Project of Jianghuai Talent Training Program in Anhui Province, Health Research Program of Anhui (No. AHWJ2023A30107) and Clinical Medicine project of Anhui Medical University.

## AUTHOR AFFILIATIONS

[1]Department of Infectious Diseases, The Second Affiliated Hospital of Anhui Medical University, Hefei, China

[2]Department of Clinical Medicine, The Second School of Clinical Medicine, Anhui Medical University, Hefei, Anhui, China

[3]Clinical Virus Research Institute, The Second Affiliated Hospital of Anhui Medical University, Hefei, China

## AUTHOR ORCIDs

Zhenhua Zhang http://orcid.org/0000-0002-8480-9004

## AUTHOR CONTRIBUTIONS

Jianmei Zhou, Conceptualization, Project administration, Supervision, Writing – original draft | Jie Zhu, Conceptualization, Data curation, Formal analysis, Software, Supervision, Validation, Visualization | Pengyue Zhang, Data curation, Formal analysis, Resources | Chunhui Tao, Formal analysis, Methodology | Xiaodan Hong, Writing – original draft, Writing – review and editing | Zhenhua Zhang, Conceptualization, Project administration, Supervision

## DATA AVAILABILITY

The data used in this study can be obtained online (http://ghdx.healthdata.org/gbd-results-tool).

## ADDITIONAL FILES

The following material is available online.

### Supplemental Material

**File S1 (Spectrum01290-24-s0001.xlsx).** Age-standardized mortality rate (ASMR) and age-standardized disability-adjusted life years rate (ASDR) for all Countries in 2019, and the average annual percent change (AAPC) of these two indicators over 30 years.
**Supplemental material (Spectrum01290-24-s0002.pdf).** Supplemental information for SDI related analysis and disease prediction.

### Open Peer Review

**PEER REVIEW HISTORY (review-history.pdf).** An accounting of the reviewer comments and feedback.

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
