## [Reviewer comments · Microbiology Spectrum]

Microbiology Spectrum

Global, regional, and national burdens of *Clostridioides difficile* infection over recent decades: a trend analysis informed by the Global Burden of Disease Study

Jianmei Zhou, Jie Zhu, Pengyue Zhang, Chunhui Tao, Xiaodan Hong, and Zhenhua Zhang

Corresponding Author(s): Zhenhua Zhang, Second Affiliated Hospital of Anhui Medical University

Review Timeline:

Submission Date:	May 30, 2024
Editorial Decision:	June 20, 2024
Revision Received:	August 6, 2024
Editorial Decision:	September 20, 2024
Revision Received:	September 28, 2024
Editorial Decision:	November 1, 2024
Revision Received:	November 29, 2024
Editorial Decision:	December 13, 2024
Revision Received:	December 18, 2024
Editorial Decision:	December 25, 2024
Revision Received:	December 29, 2024
Accepted:	February 18, 2025

Editor: Yuan Pin Hung

Reviewer(s): Disclosure of reviewer identity is with reference to reviewer comments included in decision letter(s). The following individuals involved in review of your submission have agreed to reveal their identity: Bobby G Warren (Reviewer #2)

Transaction Report:

DOI: <https://doi.org/10.1128/spectrum.01290-24>

Re: Spectrum01290-24 (**Global, regional, and national burdens of Clostridium difficile infection from 1990 to 2019: a trend analysis based on the Global Burden of Disease Study 2019**)

Dear Prof. Zhenhua Zhang:

Thank you for the privilege of reviewing your work. Below you will find my comments, instructions from the Spectrum editorial office, and the reviewer comments.

Please clearly define all the terms and describe the detail methodology in this study, including YLLs, YLD, AAPC, ASDR, ASMR, APC, BAPC

Revision Guidelines

Sincerely,
Yuan Pin Hung
Editor
Microbiology Spectrum

Reviewer #1 (Comments for the Author):

See attached

Reviewer #2 (Comments for the Author):

Summary:

The study investigates the global, regional, and national burdens of Clostridium difficile infection (CDI) from 1990 to 2019 using data from the Global Burden of Disease Study 2019. The findings reveal an overall increase in CDI burden, especially in high socio-demographic index (SDI) regions, with a notable rise in disability-adjusted life years (DALYs) and mortality rates. The burden is most significant among the elderly (over 70) and children under five. High SDI regions showed the fastest growth rates, while some low and middle SDI regions also experienced substantial burdens.

Limitations:

- Potential underreporting and underdiagnosis in regions with limited healthcare resources.
- Lack of specific estimates for CDI incidence and prevalence.
- Reliance on publicly available data which may have inherent inaccuracies or inconsistencies.

Major Concerns

- The limitations section: The authors note limitations in the appropriate section but it should be expanded. It begins with "...several limitations" but only lists two so I suspect some were removed that should be added back.
 - o Data Reliability and Completeness: The reliance on publicly available data may result in underreporting, especially in low-resource settings, potentially skewing the results. The authors note this, but it should be expanded on. The idea is that higher SDI = higher CDI, however, the comparator group would be low SDI which are also areas that would under-detect and report CDI causes a bigger gap than is likely present. The authors should describe why this impact doesn't change the conclusion. Consider citing studies that have shown a similar pattern with reliable low SDI detection.
 - o Lack of Specificity in Estimates: The study lacks specific estimates for CDI incidence and prevalence, limiting the ability to make precise conclusions. Discuss this more. How does this likely impact the data here?

Minor Issues

- Potential Bias in Data Collection: Differences in healthcare access and diagnostic practices across regions could introduce bias.
- Variation in Antibiotic Use: Differences in antibiotic prescribing practices between high and low SDI regions are not fully accounted for.
- Limited Discussion on Prevention: The study does not extensively discuss preventive measures or strategies to mitigate CDI burden and how they may differ by SDI.

Grammar Issues

- Abstract: "should also be alarmed" should be "should also be alarming."
- Introduction: "due to a toxin-producing species of Clostridium in hamsters" could be rephrased for clarity.
- Method: The phrase "and analyzed globally regionally and nationally" should include commas: "and analyzed globally, regionally, and nationally."
- Results: "We found that the overall trend indicated a heavier disease burden in regions with higher SDI with burden increasing at a faster rate" should be: "We found that the overall trend indicated a heavier disease burden in regions with higher SDI, with the burden increasing at a faster rate."

Summary:

Zhou et al. seek to characterize the global burden of *C. difficile* infection over the last three decades ending in 2019 using the data from the Global Burden of Disease Study 2019. They find that *C. difficile* burden increased through 2016 and then largely plateaued (at least in high socioeconomic index countries). They find highest colonization in ages above 70, as well as a suggested notable level of burden in children under 5 years old. While these general trends are not novel, characterization of these trends across the world is an important and under characterized aspect of the field studying *C. difficile* burden.

Major Critiques:

1. L204 The authors report “a relatively large number of deaths” in individuals <5 years old. However, the exact meaning of this is not entirely clear as the rate in individuals that young is low compared to other age groups.
 - a. Similarly, at L235, it states that children bear a “heavy burden”, but it is not clear what this means since rates and deaths in this group are small compared to other groups.
 - b. They cite a high loss in life years (DALYs), but this statistic appears to lack context. Given that the number of years lost will be greater for young children rather than adults, a comparison to other diseases with similar morbidity or mortality would be helpful. Without these caveats add to the analysis, I worry that DALYs overinflate mortality in younger individuals. For example, comparing DALYs across age groups may not be a fair comparison, but comparing DALYs within an age group would be fair. I think this is apparent since their age-standardized metrics (ASDR and ASMR) show a very modest impact on this group, if any.
 - i. Note: I am not super familiar with use of DALYs. While they are used widely and have some level of age-weighting, their use is also criticized (PMID: 10176779; PMID: 10176779). If the authors find this to be a helpful metric, its use in this context should be thoroughly justified in both introduction/methods (i.e., rationale) and discussion (i.e., limitations of DALYs as a metric in the context of this study).
 - c. Additionally, they mention in passing that diarrhea is the second leading cause of death in children under <5 as being a reason to pay attention to this group, but it is unclear if the primary cause of that diarrhea would be CDI, or something else (i.e. *Vibrio cholera*). Similarly, they cite the number of deaths but it is unclear whether these are deaths attributable to CDI (i.e., they died from something other than CDI, while having an infection). Factoring in these alternative causes of death would assist the authors in pushing home their message about the risks of *C. difficile* carriage in young individuals. Does the source dataset provide this additional information? Are there comorbidities that need to be included in the analysis?
 - i. Also, were statistics applied to these trends commented on at L204-207? If so, it is not mentioned in text or in the legend. This seems problematic.
2. It is known that infants (<12 months) have higher colonization with *C. difficile* but low or no signs of disease (PMC10311876 for example). However, the authors do not engage with this large body of literature at all. Doing so would add additional weight to their exhortation to increase surveillance among young individuals.

3. L213, 245, etc. the authors refer to epidemiological change as the primary explanation for why CDI burden changes in certain regions. However, it is very opaque what this “epidemiological change” actually is in either the methods or results. Some contexts (i.e., L213 and following) suggest this refers to the rate of infection. But then L245 says the determinant of CDI burden (i.e. infection rate?) was epidemiological change (rate?), which would be circular logic.
 - a. Please define what is meant by the epidemiological change included in the model. If it is a combination of characteristics, a deeper dive into the contributions of each of these characteristics would be helpful in informing their conclusions. If it is purely that rate of infection determines rate of death, that’s fine (but also not the most revolutionary finding).

Minor Critiques:

1. *C. difficile* should be italicized throughout the manuscript
2. Please redefine abbreviations at the start of the results section (i.e., DALY, ASDR, ASMR, etc.) for those of us who haven’t see these abbreviations often. While not necessary, it will save people like me from jumping back to the abstract a couple of times (perhaps at L145-151?)
3. L38 use of “alarmed” seems out of place here, maybe use a different word?
4. L41 “moreover, young children need to be warned” what does this look like? Is this an appropriate statement? Should it be rephrased to something like, “clinicians should take care to consider burden in individuals under 5 years of age”? Who is the audience of this manuscript?
5. L53 duplicated word “abuse, misuse”
6. L76 citations needed to further elaborate on properties of the immune system in the young and elderly
7. L158 the paragraph starts with “most developed countries... but there are exceptions.” However, those exceptions are not made abundantly clear and the reader is left to figure out which countries in the paragraph fall into that status or not.
8. L168 what does “observing from a map perspective” mean? This does not seem like a reliable way to detect differences as it is likely to miss subtle differences. Please add a supplemental figure with the map data plotted (i.e., as a box/whisker or bar plot) to assist with these conclusions.
9. L173 Table 1 is not referenced until L177 despite the data being referred to several lines earlier. I would suggest referencing that table in both places
10. L178-179 The R value listed in text vs in the figure for panels B and C do not match those listed in the text (they appear to have been swapped). Please fix based on the correct arrangement.
11. L185 Figure 4 is referred to before figure 3, perhaps reorder the figures?
12. L257 “host, environmental, bacterial, microbiome or even cultural factors” is used to hypothesize undescribed features. However, these terms are often overlapping and poorly defined (i.e., the microbiota is composed of bacteria, but also other prokaryotes and eukaryotes; the host is an environment but also cultural factors might be considered an environment). Please be more explicit in to what these terms refer to make this a meaningful list of potential explanations (rather than an overly broad laundry list).

13. L270 A number of factors that might explain why the elderly are more disposed to CDI, but no citation is given to back up these characteristics. Engagement with the primary literature would be helpful (i.e., preferably not just a review paper).
14. Fig. 3: The rise in each metric is driven by individuals over age 70, please resort the barplots so that those that change the least are at the bottom (i.e., younger people) so it is easier to visually compare changes in these groups of individuals.

Open Data Request:

1. L141 If you could publish your code to a public database, such as GitHub or an equivalent, that would be incredibly helpful to the field! Particularly given ASM's commitment to open data science, this would greatly aid in increased data reproducibility and transparency.

August 1, 2024

Dear editors and reviewers,

Thank you for the opportunity to revise our manuscript (ID: Spectrum01290-24) entitled "**Global, regional, and national burdens of Clostridium difficile infection from 1990 to 2019: a trend analysis based on the Global Burden of Disease Study 2019**" We appreciate the detailed feedback provided by the reviewers and the editorial team. We have carefully considered all the comments and have made the necessary revisions to address each point raised. Below, we provide a **point-by-point** response to the reviewers' comments and made the relevant revisions in the manuscript.

Editor

Please clearly define all the terms and describe the detail methodology in this study, including YLLs, YLD, AAPC, ASDR, ASMR, APC, BAPC.

Our response:

- YLLs: The years of life lost (YLLs) refer to the number of years of expected life lost due to premature death, calculated by multiplying cause-age-sex-location-year-specific deaths by the standard life expectancy at the age that death occurred (PMID: 33069326).
- YLD: The years lived with disability (YLD) refer to the number of healthy years lost due to poor health or disability, calculated by multiplying the cause-age-sex-location-year-specific prevalence of sequelae by their respective disability weights for each disease and injury. Disability weight for each cause describes the severity of health loss with that specific cause, ranging from 0 (full health) to 1 (death) (PMID: 33069326).
- APC and AAPC: APC (annual percent change) represents the annual percentage change in a particular metric (e.g., DALYs, mortality) over a specified period, reflecting the annual trend in the ratio. AAPC (average annual percentage changes) refers to the average annual percentage change across multiple time periods, used to describe the overall trend over a longer period of time. It is

calculated by fitting a regression line to the natural logarithm of the rates and provides a summary measure of trend over time.

- ASDR: By standardizing to a common age structure, ASDR (age-standardized DALYs rate) allows for meaningful comparisons of DALYs between populations with varying age demographics.
- ASMR: ASMR (age-standardized mortality rate) is a mortality rate that has been adjusted to a standard age distribution. This adjustment allows for fair comparisons between populations with different age structures by eliminating the effects of age distribution differences.
- BAPC: BAPC (Bayesian age-period-cohort) is an advanced statistical method that predicts disease DALYs rate and mortality by accounting for the effects of age, time period, and birth cohort, as well as their interactions. Utilizing Bayesian statistical methods, the model incorporates prior information to flexibly model and estimate uncertainty. By integrating observed data on age, time period, and birth cohort, the BAPC model captures how these factors influence disease incidence rates and forecasts future trends.

Reviewer #1 (Comments for the Author):

1. L204 The authors report “a relatively large number of deaths” in individuals <5 years old. However, the exact meaning of this is not entirely clear as the rate in individuals that young is low compared to other age groups.

- a. Similarly, at L235, it states that children bear a “heavy burden”, but it is not clear what this means since rates and deaths in this group are small compared to other groups.

Our response:

This is indeed a very valuable reminder, and we apologize for any inappropriate phrasing in our previous statements. Based on our experience, CDI is more common in the elderly population. However, through analyzing different age groups, we found

that children under 5 years old had the highest number of deaths and DALYs within this vulnerable group of children. Additionally, the number of deaths in children under 5 years old exceeded that of all other age groups under 45 years old. For an opportunistic pathogen and antibiotic-associated infection like CDI, enhancing prevention and treatment measures could significantly reduce the DALY rates and mortality rates in children, which is highly meaningful. Therefore, our goal is to increase awareness and attention to pediatric CDI among clinicians and public health professionals.

In the manuscript, we revised statements like "a relatively large number of deaths" and "the heavy burden on children under 5" to "although their DALYs rates and mortality were relatively low, this age group had the highest number of deaths and DALYs within this vulnerable group of children (< 14 years)($P < 0.0001$). Additionally, the number of deaths in children under 5 years old exceeded that of all other age groups under 45 years old ($P < 0.0001$). There is a need to gradually increase awareness and implement targeted prevention and treatment measures.

b. They cite a high loss in life years (DALYs), but this statistic appears to lack context. Given that the number of years lost will be greater for young children rather than adults, a comparison to other diseases with similar morbidity or mortality would be helpful. Without these caveats add to the analysis, I worry that DALYs overinflate mortality in younger individuals. For example, comparing DALYs across age groups may not be a fair comparison, but comparing DALYs within an age group would be fair. I think this is apparent since their agestandardized metrics (ASDR and ASMR) show a very modest impact on this group, if any.

i. Note: I am not super familiar with use of DALYs. While they are used widely and have some level of age-weighting, their use is also criticized (PMID: 10176779; PMID: 10176779). If the authors find this to be a helpful metric, its use in this context should be thoroughly justified in both introduction/methods (i.e., rationale) and

discussion (i.e., limitations of DALYs as a metric in the context of this study).

Our response:

Let's elaborate on the current GBD database's calculation method for DALYs (disability-adjusted life years). The formula for calculating DALYs is: $DALYs = YLLs + YLDs$, YLLs (years of life lost) are calculated by multiplying cause-age-sex-location-year-specific deaths by the standard life expectancy at the age that death occurred. This represents the years of life lost due to premature mortality; YLDs (years lived with disability) are calculated by multiplying the cause-age-sex-location-year-specific prevalence of sequelae by their respective disability weights for each disease and injury.

Indeed, comparing DALYs between groups with large age differences is not entirely fair, but presenting DALYs by age group has its corresponding value. Some top-tier articles also stratify age into five-year intervals (PMID: 38642570). Under the same number of disabilities, the DALYs calculated for younger populations should be higher than for older groups. However, our study data indicate that the DALYs in younger groups are significantly lower than in older groups, which further suggests that the burden of CDI in young people is significantly lower. Moreover, the DALYs between two age groups with close ages are reasonably comparable. For example, the DALYs for children under five years old are significantly higher than those for the 5-10 year age group, which is a more pronounced difference than between other adjacent groups. This can effectively highlight the severity in the under-five population.

Thank you for reminding us to consider critical literature on DALYs. The critique in PMID: 10176779 mainly focuses on the social and ethical issues brought by age weighting and disability weight (DW) allocation, as well as the problems of social resource distribution and discrimination caused by time preference and future life discounting. However, GBD2019 has removed these calculation methods, currently considering health losses across all ages and time periods equally important. This aims to assess the health burden more fairly, regardless of when the health loss occurs. Despite this, every evaluation metric has its drawbacks and its value. The

original purpose of age-weighted DALYs was to emphasize the greater impact of death or disability in younger age groups on social productivity. However, when using non-age-weighted DALYs, we can indirectly demonstrate the objective loss of social productivity due to CDI by stratifying the data by age groups.

Lastly, the analysis in Figure 2 not only shows the burden across different age groups but also illustrates the trend over 30 years for these age groups. Your feedback has been very helpful in improving the readability of the article. We have added the improvements and more detailed calculations and descriptions of GBD2019's DALYs in the methods and discussion sections.

c. Additionally, they mention in passing that diarrhea is the second leading cause of death in children under <5 as being a reason to pay attention to this group, but it is unclear if the primary cause of that diarrhea would be CDI, or something else (i.e. *Vibrio cholera*). Similarly, they cite the number of deaths but it is unclear whether these are deaths attributable to CDI (i.e., they died from something other than CDI, while having an infection). Factoring in these alternative causes of death would assist the authors in pushing home their message about the risks of *C. difficile* carriage in young individuals. Does the source dataset provide this additional information? Are there comorbidities that need to be included in the analysis?

i. Also, were statistics applied to these trends commented on at L204-207? If so, it is not mentioned in text or in the legend. This seems problematic.

Our response:

1) Clarification on the Causes of Fatal Diarrhea:

In our manuscript, we mention that diarrhea is the second leading cause of death among children under five to emphasize the vulnerability of this age group. We did not imply that CDI is the primary cause. According to the literature, rotavirus, *Shigella* species, and other pathogens are the major causes of fatal diarrhea in children (PMID:

36266651, PMID: 24023773, PMID: 32115003). Although the causative pathogens differ, there may be common mechanisms related to diarrhea, such as the underdeveloped immune system in children, immature intestinal barriers, and poor tolerance to dehydration. Furthermore, our data indicate that while the case-fatality rate and disability rate associated with CDI-related diarrhea in children under five are relatively low, there has been an overall upward trend in recent years. Our aim is to raise awareness within the public health system about the importance of this age group.

2) Clarification on the Attribution of Mortality:

The GBD database does not simply categorize and display collected data. Instead, it undergoes a rigorous process of data review and calibration, involving scientific literature, discharge records (from multiple countries), inpatient records, outpatient data, and insurance claims, conducted through international expert consultations and global surveys. Based on this, the data is further processed, including the application of standardized computational methods and diagnostic criteria to minimize errors and enhance comparability. Additionally, the data and methodologies are regularly updated to reflect the latest scientific research and data collection outcomes, ensuring continuous correction and improvement of the data. Therefore, the CDI-related diarrhea mortality data presented by GBD represents the processed and recognized global mortality or disability data attributable to CDI-related diarrhea. This data does not include other disease-related characteristics, covariates, or comorbidities.

The confirmation of mortality attribution in the data processing is achieved through a series of mathematical algorithms used to construct models, including three main standardized tools: cause of death ensemble model (CODEm), DisMod-MR, and spatiotemporal Gaussian process regression (ST-GPR). Briefly, CODEm is a highly systematized tool for analyzing cause of death data using an ensemble of different modeling methods for rates or cause fractions with varying choices of covariates that perform best with out-of-sample predictive validity testing. DisMod-MR is a Bayesian meta-regression tool that allows evaluation of all available

data on incidence, prevalence, remission, and mortality for a disease, enforcing consistency between epidemiological parameters. ST-GPR is a set of regression methods that borrow strength between locations and over time for single metrics of interest, such as risk factor exposure or mortality rates. Additionally, for select diseases, particularly for rarer outcomes, alternative modeling strategies have been developed, which are described in Appendix 1 (Section 3.2 of PMID:33069326).

Regarding diseases like diarrhea (including CDI-associated diarrhea), the details of data collection and construction are as follows (see figure below). We have added the above description to the background and discussion sections.

3) Clarification on statistics

Thank you for your reminder. We had indeed not performed a statistical analysis of the trends commented on in lines 204-207 previously. Therefore, we firstly compared the differences in number of deaths between < 5 age group and other age groups using the paired rank-sum test by year, and the results showed statistical significance ($P < 0.0001$). Additionally, to ensure fairness, we conducted a paired rank-sum test on the differences in DALYs between adjacent age groups (< 5 years subtraction 5-9 years vs 5-9 years subtraction 10-14 years). The results indicated significant differences ($P < 0.0001$). This showed that among children under 14 years old, the <5 age group has the highest number of DALYs and deaths. We have added the results to the main text.

2. It is known that infants (<12 months) have higher colonization with *C. difficile* but low or no signs of disease (PMC10311876 for example). However, the authors do not engage with this large body of literature at all. Doing so would add additional weight to their exhortation to increase surveillance among young individuals.

Our response:

Thank you for your insightful suggestion. We agree that engaging with the literature on the high colonization rates of *C. difficile* in infants with low or no signs of disease will add valuable context to our discussion. We will incorporate this important aspect into our manuscript to strengthen our argument for increased surveillance among young individuals. Specifically, we will include references such as PMC10311876 and others (PMC8600012, PMC10471512) to provide a comprehensive overview of this phenomenon. We will include the following content in the manuscript:

It is noteworthy that although the colonization rate of *C. difficile* in infants is high, clinical infections are rare. It has been reported that the colonization rate is about 37% in the first month after birth and decreases to around 30% between 1 to 6 months of age. The rate continues to decline, reaching about 10% in healthy infants by 1 year of age. By 3 years of age, the colonization rate is approximately 3%, similar

to the rate in adult carriers (PMID: 7244558, PMID: 15906260, PMID: 20512057). The resistance of newborns to CDI may be related to the absence of toxin receptors in their immature intestinal mucosa, the lack of downstream signaling pathways, and certain protective factors found in breast milk and the host gut microbiome (PMID: 31253983, PMID: 1325998). However, CDI may manifest in infants with congenital megacolon (PMID: 7888555). Additionally, a review study (PMID: 23460123) has indicated that exposure to antibiotics, a history of hospitalization, solid organ transplantation, the presence of a gastrostomy or jejunostomy tube, and underlying comorbidities such as cardiovascular disease, cancer, and inflammatory bowel disease (IBD) are risk factors for CDI in children. Therefore, special attention should be given to these populations.

3. L213, 245, etc. the authors refer to epidemiological change as the primary explanation for why CDI burden changes in certain regions. However, it is very opaque what this “epidemiological change” actually is in either the methods or results. Some contexts (i.e., L213 and following) suggest this refers to the rate of infection. But then L245 says the determinant of CDI burden (i.e. infection rate?) was epidemiological change (rate?), which would be circular logic.

a. Please define what is meant by the epidemiological change included in the model. If it is a combination of characteristics, a deeper dive into the contributions of each of these characteristics would be helpful in informing their conclusions. If it is purely that rate of infection determines rate of death, that’s fine (but also not the most revolutionary finding).

Our response:

Thank you for your insightful comments regarding the definition of “epidemiological change” in our study.

First, the main drivers behind changes in disease burden are threefold: age structure, population growth, and epidemiological changes. Each factor affects disease burden differently across regions and time periods. The purpose of the decomposition analysis model is to determine which factor is the primary driver of

changes in disease burden. Given that the preliminary conclusions of this study indicate a higher incidence rate among the elderly, our decomposition analysis allows us to observe the weight of aging or population structure on these results. This will more clearly highlight the primary driving factors. Our study indicates that in high SDI regions, the changes in CDI burden are mainly driven by epidemiological changes. These epidemiological changes may be due to the prevalence of highly virulent strains, antibiotic exposure, increased healthcare access, and climate change in high SDI regions.

Secondly, within the framework of the decomposition analysis model, "epidemiological change" refer to shifts in disease incidence, prevalence, and mortality rates, influenced by various factors such as pathogen variations (e.g., the emergence of highly virulent strains), the effectiveness of public health interventions (e.g., vaccination and antibiotic use), lifestyle changes (e.g., diet and exercise habits), environmental factors (e.g., climate change), and socioeconomic changes (e.g., access to and quality of healthcare resources). Epidemiological changes affect the spread and severity of diseases, thereby altering the disease burden.

Lastly, the logic of "infection rate determining infection rate" is a misunderstanding. In fact, we are not suggesting that "infection rate determines infection rate," but rather that changes in infection rates (as part of epidemiological change) influence disease burden. Here, "epidemiological change" encompasses not only simple changes in infection rates but also includes multiple factors that affect infection rates. In the Methods section, we explicitly defined "epidemiological change" as the combined influence of multiple factors. In the discussion section of our manuscript, we analyzed how these epidemiological changes impact disease burden.

Minor Critiques:

1. *C. difficile* should be italicized throughout the manuscript

Our response:

Thank you for pointing this out. We have revised the manuscript to italicize "*C. difficile*" throughout the text.

2. Please redefine abbreviations at the start of the results section (i.e., DALY, ASDR, ASMR, etc.) for those of us who haven't see these abbreviations often. While not necessary, it will save people like me from jumping back to the abstract a couple of times (perhaps at L145-151?)

Our response:

Thank you for the suggestion. We have redefined the abbreviations (DALY, ASDR, ASMR, etc.) at the start of the results section for clarity and convenience, as recommended.

The global burden of *Clostridium difficile* infection (CDI) was assessed using various metrics, including disability-adjusted life years (DALYs), age-standardized disability rates (ASDR), age-standardized mortality rates (ASMR), average annual percentage changes (AAPCs) and socio-demographic index (SDI).

3. L38 use of "alarmed" seems out of place here, maybe use a different word?

Our response:

Thank you for your suggestion. We have revised the sentence to replace "alarmed" with "considered" for better clarity and appropriateness.

4. L41 "moreover, young children need to be warned" what does this look like? Is this an appropriate statement? Should it be rephrased to something like, "clinicians should take care to consider burden in individuals under 5 years of age"? Who is the audience of this manuscript?

Our response:

Thank you for pointing this out. We apologize for the inappropriate phrasing. We have revised the sentence to: "Moreover, clinicians should take care to consider the burden in individuals under 5 years of age." This change ensures clarity and appropriateness for our target audience. We appreciate your feedback and have made the necessary adjustments accordingly.

5. L53 duplicated word "abuse, misuse"

Our response:

Thank you for highlighting this issue. We have removed the duplicated word “misuse” in the sentence. This adjustment improves clarity and eliminates redundancy in our manuscript.

6. L76 citations needed to further elaborate on properties of the immune system in the young and elderly.

Our response:

Thank you for your feedback. We will incorporate additional citations to further elaborate on the properties of the immune system in both young and elderly populations, ensuring comprehensive coverage of this aspect in our manuscript. I will add the following content: Elderly individuals are more susceptible to infections, including CDI, due to immunosenescence, which involves reduced T cell and B cell function, increased inflammatory responses, and diminished response to new antigens (PMID: 29242543). Children, especially infants and newborns, have immature immune systems, making them more susceptible to infections like CDI. Their gut microbiota and immune systems develop gradually after birth, influenced by factors such as mode of delivery, feeding practices, and environmental exposures (PMID: 26702035, PMID: 27126036).

7. L158 the paragraph starts with “most developed countries... but there are exceptions.” However, those exceptions are not made abundantly clear and the reader is left to figure out which countries in the paragraph fall into that status or not.

Our response:

Thank you for your suggestion. We have clarified the exceptions in the paragraph by explicitly listing the countries that do not follow the general trend, specifically mentioning South Africa and Venezuela. Revised: Nationally, most developed countries have greater CDI burden than developing countries, but there are exceptions, such as South Africa and Venezuela, where the CDI burden were also

significant.

8. L168 what does “observing from a map perspective” mean? This does not seem like a reliable way to detect differences as it is likely to miss subtle differences. Please add a supplemental figure with the map data plotted (i.e., as a box/whisker or bar plot) to assist with these conclusions.

Our response:

Thank you for your valuable feedback. We understand that observing data from a map perspective may overlook some subtle differences. However, the world map illustrates the disease burden of *C. difficile* in 204 countries, making it challenging to create box plots or bar charts. Nevertheless, we can provide an Excel sheet containing specific data for all 204 countries to further enhance data reliability (See Supplementary File1 of revised manuscript).

9. L173 Table 1 is not referenced until L177 despite the data being referred to several lines earlier. I would suggest referencing that table in both places

Our response:

Thank you for your suggestion. We have revised the manuscript to reference Table 1 at both L173 and L177, ensuring the data is properly cited and easy to locate for the reader. We appreciate your careful review and feedback.

10. L178-179 The R value listed in text vs in the figure for panels B and C do not match those listed in the text (they appear to have been swapped). Please fix based on the correct arrangement

Our response:

Thank you for pointing out this error. We have reviewed the R values and corrected the arrangement in the text to match the figures for panels B and C. We appreciate your careful review and feedback.

11. L185 Figure 4 is referred to before figure 3, perhaps reorder the figures?

Our response:

Thank you for your suggestion. We will reorder the figures so that Figure 3 precedes Figure 4 to align with their references in the text. This adjustment will ensure clarity and coherence in presenting the figures. We appreciate your attention to detail and feedback.

12. L257 “host, environmental, bacterial, microbiome or even cultural factors” is used to hypothesize undescribed features. However, these terms are often overlapping and poorly defined (i.e., the microbiota is composed of bacteria, but also other prokaryotes and eukaryotes; the host is an environment but also cultural factors might be considered an environment). Please be more explicit in to what these terms refer to make this a meaningful list of potential explanations (rather than an overly broad laundry list).

Our response:

Thank you for your suggestions. In explaining the reasons for the lower CDI burden in low SDI regions, I indeed used some overly general terms, some of which may overlap and lack clear definitions. We have now provided a detailed explanation for the lower CDI burden in low SDI regions and revised the manuscript as follows:

However, the lower CDI burden in low SDI regions may be due to multiple factors. These factors include the abundance of gut microbiota, dietary structure, and the virulence of prevalent strains, which interact with each other. Studies have shown that the abundance of *Prevotella* and *Bacteroides* in the gut microbiota of children in Africa is significantly higher than in European children, and these two gut bacteria can protect the body and reduce CDI (PMID: 24803517, PMID: 22972295). Due to the impact of economic and agricultural development levels, as well as dietary cultural practices, the dietary structure in low SDI regions has a higher proportion of fiber (PMID: 0679230), which helps increase the abundance of gut microbiota. Additionally, although there is a problem of low-grade antibiotic misuse in low SDI countries, the overall use of broad-spectrum, high-grade antibiotics is still lower than in high SDI countries (PMID: 26603918), and reports of highly virulent strains are mainly

concentrated in high SDI regions (PMID: 16965399).

13. L270 A number of factors that might explain why the elderly are more disposed to CDI, but no citation is given to back up these characteristics. Engagement with the primary literature would be helpful (i.e., preferably not just a review paper).

Our response:

Thank you for your suggestion. In the manuscript, I have provided a more detailed explanation of the reasons why elderly individuals are more susceptible to CDI, supported by several references. The revised content is as follows:

Studies have shown that the high prevalence of CDI is closely related to dysbiosis of the gut microbiota (PMID: 27158839, PMID: 37318134). Elderly individuals generally have lower overall gut microbiota abundance and lower abundance of probiotics in medical settings (PMID: 22797518, PMID: 33764826). Additionally, gut microbiota imbalance is often caused by factors such as nutritional deficiencies following acute or chronic diseases, long-term and extensive use of antibiotics, and abdominal surgeries (PMID: 18043614, PMID: 25250176), conditions that are more common among the elderly. Furthermore, the immune system of elderly hosts responds more slowly to microbial molecules that cross the epithelial barrier, leading to rapid development of *Clostridioides difficile* toxin-mediated colonic epithelial damage. The loss of epithelial integrity results in increased intestinal permeability, allowing bacteria to translocate from the gut lumen to deeper tissues (PMID: 22797518). Research indicates that the severity of the disease is negatively correlated with the presence of toxin-specific immunoglobulin A (IgA) and IgG antibodies (PMID: 1431247, PMID: 10666429). Due to the poorer immune system function in the elderly, they produce fewer specific antibodies, making them more susceptible to infection.

14. Fig. 3: The rise in each metric is driven by individuals over age 70, please resort the barplots so that those that change the least are at the bottom (i.e.,

younger people) so it is easier to visually compare changes in these groups of individuals.

Our response:

Thank you for your valuable feedback. We will revise Figure 3 to reorder the bar plots, placing the age groups that change the least at the bottom (i.e., younger people) to make it easier to visually compare the changes among different age groups. This adjustment should enhance the clarity and readability of the figure.

Reviewer 2

Summary:

The study investigates the global, regional, and national burdens of Clostridium difficile infection (CDI) from 1990 to 2019 using data from the Global Burden of Disease Study 2019. The findings reveal an overall increase in CDI burden, especially in high socio-demographic index (SDI) regions, with a notable rise in disability-adjusted life years (DALYs) and mortality rates. The burden is most significant among the elderly (over 70) and children under five. High SDI regions showed the fastest growth rates, while some low and middle SDI regions also experienced substantial burdens.

Limitations:

- Potential underreporting and underdiagnosis in regions with limited healthcare resources.
- Lack of specific estimates for CDI incidence and prevalence.
- Reliance on publicly available data which may have inherent inaccuracies or inconsistencies.

Major Concerns

- The limitations section: The authors note limitations in the appropriate section but it should be expanded. It begins with "...several limitations" but only lists two so I suspect some were removed that should be added back.

Our response:

We appreciate the reviewer's valuable feedback. We agree that the "limitations" section should be expanded, and we will include the previously omitted content in the manuscript. We will include the following content in the manuscript:

Additionally, the GBD data does not differentiate between the severity of CDI cases, nor can it distinctly separate the burden of hospital-acquired infections from community-acquired infections. Lastly, we could not explore the causal relationship between antibiotic use and the burden of CDI in detail, as the GBD data does not include specific prescription information. Therefore, future research should integrate

multiple data sources and more detailed analyses to better understand the global burden and influencing factors of CDI.

o **Data Reliability and Completeness:** The reliance on publicly available data may result in underreporting, especially in low-resource settings, potentially skewing the results. The authors note this, but it should be expanded on. The idea is that higher SDI = higher CDI, however, the comparator group would be low SDI which are also areas that would under-detect and report CDI causes a bigger gap than is likely present. The authors should describe why this impact doesn't change the conclusion. Consider citing studies that have shown a similar pattern with reliable low SDI detection.

Our response:

Thank you for your valuable feedback on our article. We indeed recognize that relying on publicly available data may lead to underreporting in low-resource areas, potentially affecting the accuracy of the results. Specifically, low SDI regions may have insufficient detection and reporting of CDI cases, making the gap between high SDI and low SDI regions appear larger. To address this issue, we have expanded the discussion on this limitation in our manuscript.

Firstly, the underreporting of CDI in low SDI countries and regions is indeed a concern. The main reasons are as follows:

- **Lack of awareness:** Diarrhea symptoms are extremely common, and most cases are self-limiting. It is challenging to recognize CDI infection unless symptoms persist or lead to severe complications, especially in community settings (PMID: 23816346).
- **Limited detection resources:** Even when typical symptoms occur, it may be difficult to fully implement CDI testing. This situation is more severe in areas with limited medical personnel and testing resources (PMID: 24669194).
- **Strain variation:** There is variation in the epidemic strains of CDI, with different subtypes showing distinct epidemiological and virulence characteristics in different regions. Beyond routine screening methods, targeted identification of

strain subtypes is essential for assessing disease severity and conducting epidemiological research, but this poses a greater challenge for low SDI countries (PMID: 23816346, PMID: 29368667, PMID: 33751421).

Despite these concerns, we believe this impact does not alter the overall trends and conclusions of our study for the following reasons:

- GBD2019 dataset adjusts for underreporting by using statistical models and supplementary data to mitigate this impact as much as possible. While this cannot completely eliminate the issue, these adjustments enhance the reliability of the data.
- High-risk populations: Severe CDI-associated diarrhea is more common in patients who have undergone high colonic resection, those with IBD, and those with severe infections treated with high-dose broad-spectrum antibiotics. These patients are more frequently seen in developed countries (PMID: 24282348).
- Nosocomial infections: CDI-related diarrhea is a typical nosocomial infection (PMID: 35457076). Reports indicate that CDI is among the common healthcare-associated infections in Europe (PMID: 27755545), making it more prevalent and leading to adverse outcomes in high SDI countries with better healthcare systems.
- Antibiotic use: Although antibiotic management may be relatively chaotic in low SDI countries, making antibiotics more accessible (PMID: 25667187), these antibiotics are often of lower grades, leading to asymptomatic CDI infections. Severe or fatal outcomes are less influenced by such antibiotics (PMID: 33572240, PMID: 36196441). High-grade broad-spectrum antibiotics, which are more closely associated with CDI, are predominantly used in high SDI countries (PMID: 29581252).
- Detection capability: Studies show that 50% of *Clostridium difficile* testing in the United States uses PCR nucleic acid testing. Some experts believe toxin enzyme immunoassays are more accurate. In India, the most common diagnostic methods include anaerobic culture, toxin culture, and enzyme-linked

immunosorbent assays (PMID: 32104159). Therefore, we can infer that low SDI regions are capable of detecting *Clostridium difficile* (PMID: 34104883).

- Gut microbiota: Essentially, CDI is closely related to the ecological imbalance of gut microbiota. Studies indicate that the gut microbiota ecology is better in low SDI countries (PMID: 25892234), while industrialized countries have gut microbiota influenced by multiple factors (PMID: 31089293).

In conclusion, while the overall prevalence in low SDI regions may be underestimated due to diagnostic and conceptual reasons, our study's conclusions regarding the trends of mortality and disability due to this disease remain reliable, we have expanded on this in the manuscript discussion section.

o Lack of Specificity in Estimates: The study lacks specific estimates for CDI incidence and prevalence, limiting the ability to make precise conclusions.

Discuss this more. How does this likely impact the data here?

Our response:

- Thank you for your insightful comment regarding the lack of specific estimates for CDI incidence and prevalence. We acknowledge that the absence of these data does limit the precision of our conclusions, and we address this limitation as follows:
- Comprehensive burden assessment: The lack of incidence and prevalence data means that our study primarily relies on mortality and DALY rates to estimate the burden of CDI. This limitation restricts our ability to provide a full picture of the disease burden. Incidence data would allow us to quantify the rate of new infections, while prevalence data would show the total number of existing cases at a given time. Together, these metrics are crucial for understanding the true impact of CDI on public health.
- Understanding disease dynamics: Without incidence and prevalence estimates, we may miss important epidemiological trends and patterns in CDI. For example, CDI could have a high incidence but a low mortality rate, suggesting that while many people are affected, the majority recover. Conversely, a low incidence but

high mortality rate might indicate underdiagnosis or insufficient treatment in that region. On the other hand, a high prevalence but low incidence might indicate a chronic condition with significant long-term health impacts. These distinctions are critical for tailoring public health interventions and understanding the full burden of disease.

- Policy and resource allocation: Public health policies and resource allocation decisions rely on accurate and comprehensive data. The absence of incidence and prevalence estimates could lead to misinformed decisions. For example, resources might be disproportionately allocated to managing CDI-related mortality rather than preventing new infections or addressing chronic cases. This could reduce the overall effectiveness of public health strategies aimed at controlling CDI.
- Intervention effectiveness: Evaluating the effectiveness of interventions, such as infection control measures or antibiotic stewardship programs, requires detailed incidence and prevalence data. Without these estimates, it is challenging to assess whether interventions are successfully reducing new cases and controlling the spread of CDI.
- Comparative analysis: The lack of specific incidence and prevalence data also limits our ability to compare CDI burden across different regions and populations. Such comparisons are vital for identifying high-risk groups and areas requiring targeted interventions. This limitation reduces the granularity of our analysis and may obscure important epidemiological insights.

It is important to note that the lack of incidence and prevalence data is due to the limitations of the GBD 2019 database we used, rather than a reluctance on our part to conduct such analyses. The database itself does not provide these specific estimates, which constrains the scope of our analysis. We have discussed this limitation in the manuscript's limitations section.

Minor Issues

1. Potential Bias in Data Collection: Differences in healthcare access and

diagnostic practices across regions could introduce bias.

Our response:

The GBD database does not simply present data collected from various regions directly. Instead, it gathers data from multiple sources and applies a series of complex modeling techniques to ultimately produce estimates of the burden for various diseases. The methodology and data processing of GBD take into account the heterogeneity of data sources, such as differences in healthcare access and diagnostic practices across regions. Through GBD's series of modeling methods, data inconsistencies are corrected, allowing for the analysis and comparison of diseases across different regions. The GBD study is widely recognized and supported by international health organizations and experts. Its methods and data processing have undergone rigorous peer review and evaluation by international experts, ensuring scientific rigor and reliability. Despite potential biases that may have existed during data collection, the use of comprehensive methodologies and rigorous adjustments in the GBD database has minimized these biases to the greatest extent possible. Consequently, the conclusions drawn from this analysis remain reliable and robust.

2. Variation in Antibiotic Use: Differences in antibiotic prescribing practices between high and low SDI regions are not fully accounted for.

Our response:

Thank you for your suggestion. We acknowledge that the differences in antibiotic prescribing practices between high and low SDI regions were not fully addressed. We will incorporate additional information and analysis on these variations to provide a more comprehensive understanding in the revised manuscript. The revised content is as follows:

Studies have shown that although low-income countries have poor regulation of antibiotics and increasing usage rates, the antibiotics that are more readily available in these regions are often lower-tier (PMID: 26603918), which have a lower probability of inducing severe *C. difficile* infections. Additionally, some low-SDI

countries cannot access even the most basic antibiotics and lack sufficient doctors to prescribe them (PMID: 26603919). Therefore, despite chaotic antibiotic management and rising consumption in these regions, the overall usage level remains low. In contrast, high-SDI countries have strict and regulated antibiotic usage policies, but the problem of overuse persists, especially with higher-tier broad-spectrum antibiotics like carbapenems and polymyxins (PMID: 29581252). This is likely driven by the identification of resistant strains, prompting the choice of more advanced antibiotics (PMID: 24252483). These differences in antibiotic prescribing practices between high-SDI and low-SDI regions partially explain the heavier CDI burden in high-SDI areas.

3. Limited Discussion on Prevention: The study does not extensively discuss preventive measures or strategies to mitigate CDI burden and how they may differ by SDI.

Our response:

Thank you for your suggestion. We acknowledge that the discussion on preventive measures and strategies to mitigate CDI burden was limited. We will expand this section to include a more comprehensive analysis of preventive measures and how they may differ by SDI. This will provide a clearer understanding of the strategies that could be implemented in different regions to effectively reduce the CDI burden. The revised content is as follows:

For high SDI countries, although they possess advanced healthcare systems, healthcare-associated infections remain a significant route for *C. difficile* infection (PMID: 19769680). Therefore, it is necessary to control patient density, promptly detect high-risk pathogens, and strengthen isolation, disinfection, and hand hygiene measures. A sample survey conducted in multiple hospitals in the United States showed that from 2011 to 2015, overall healthcare-associated infection rates declined through enhanced infection control measures, but *C. difficile* infections did not significantly improve. This may be due to antibiotic use being a major driver of *C.*

difficile infection and antibiotic resistance (PMID: 18971494, PMID: 30380384). Hence, it is crucial to continue improving antibiotic prescription practices, such as implementing strict antibiotic stewardship programs, enhancing education and training for healthcare personnel, and strengthening the monitoring and analysis of antibiotic usage data.

For low SDI regions, besides basic measures such as hand washing, contact isolation, and environmental cleaning (PMID: 28779831), it is also necessary to develop and provide cost-effective *C. difficile* diagnostic tests, formulate reasonable antibiotic management plans, and minimize unnecessary and over-the-counter antibiotic use. These measures will help effectively reduce the burden of CDI in these regions.

Grammar Issues

1. Abstract: "should also be alarmed" should be "should also be alarming."

Our response:

Thank you for your suggestion. We will revise the abstract to replace "should also be alarmed" with "should also be considered," as we believe this change will better convey the intended meaning.

2. Introduction: "due to a toxin-producing species of *Clostridium* in hamsters" could be rephrased for clarity.

Our response:

Thank you for your comment. Upon reviewing our manuscript, it appears there is a misunderstanding. The sentence "due to a toxin-producing species of *Clostridium* in hamsters" is actually the title of the reference cited ("*C. difficile* was first identified as the pathogen associated with antibiotic-related diarrhea in the late 1970s"). This sentence is not part of our Introduction section. If there are specific concerns or areas where clarity is needed within the Introduction, please let us know so we can address them accordingly.

3. Method: The phrase "and analyzed globally regionally and nationally" should

include commas: "and analyzed globally, regionally, and nationally."

Our response:

Thank you for your suggestion. We will revise the phrase to include commas as follows: "and analyzed globally, regionally, and nationally."

4. Results: "We found that the overall trend indicated a heavier disease burden in regions with higher SDI with burden increasing at a faster rate" should be: "We found that the overall trend indicated a heavier disease burden in regions with higher SDI, with the burden increasing at a faster rate."

Our response:

Thank you for your suggestion. We will revise the sentence as follows: "We found that the overall trend indicated a heavier disease burden in regions with higher SDI, with the burden increasing at a faster rate."

Thank you again for your effort in our article, and thank you very much for your consideration. We hope that the revision is acceptable and look forward to hearing from you soon. Please contact us for any questions.

With kind regard,

Yours sincerely,

Zhenhua Zhang Prof

Department of Infectious Diseases, Clinical Virus Research Institute, The Second Affiliated Hospital of Anhui Medical University.

Address: Furong Road 678, Hefei 230601, Anhui, China.

Email: zzh1974cn@163.com

Re: Spectrum01290-24R1 (**Global, regional, and national burdens of Clostridium difficile infection from 1990 to 2019: a trend analysis based on the Global Burden of Disease Study 2019**)

Dear Prof. Zhenhua Zhang:

Thank you for the privilege of reviewing your work. Below you will find my comments, instructions from the Spectrum editorial office, and the reviewer comments.

Revision Guidelines

Sincerely,
Yuan Pin Hung
Editor
Microbiology Spectrum

Reviewer #1 (Comments for the Author):

Excellent work on the revisions, all of my concerns have been addressed! I think this will be a valuable contribution to the field.

Reviewer #3 (Comments for the Author):

Dear authors,

Many thanks for the opportunity to review this interesting manuscript.

A lot of my thoughts have been put forward by the two other two referees.

However, I would be in particular cautious with the high disease burden in children since I did not encounter this aspect, neither in the literature nor in my career.

Clostridium difficile is now called *Clostridioides difficile*. This has to be corrected throughout the whole text (also the title).

Has there been any inclusion or exclusion criteria for included data? Furthermore, I do not understand why the data inclusion ended 2019. Are no newer data available.

A lot of different development took place in the meantime e.g. the COVID-19 pandemic and a decline of nosocomial/"hypervirulent" *C. difficile* strains.

Table 1 includes data for certain regions I would emphasize to make an order according to continents rather than alphabetical.

What has also not been discussed is the impact of the before mentioned strains such as ribotype 027 (RT027) and corresponding antimicrobial resistance.

With the best regards

August 23rd, 2024

Dear Dr. Yuan Pin Hung,

I am very pleased to accept your kind invitation to write this review, and very grateful for giving me this chance. Please find the attached comments for this manuscript (**ID: Spectrum01290-24R1**) entitled “**Global, regional, and national burdens of *Clostridium difficile* infection from 1990 to 2019: a trend analysis based on the Global Burden of Disease Study 2019**” by Jianmei Zhou, Jie Zhu, Pengyue Zhang, Chunhui Tao, Xiaodan Hong, Zhenhua Zhang

The objective of this study was to track the global incidence of CDI severity during the last 30 years, from 1990 to 2019, through analysis of various characteristics, including the socio-demographic index (SDI). The average annual percentage changes (AAPCs), mortality rate (ASMR), and age-standardized DALY rate (ASDR) were among the metrics used. The CDC reports that *C. difficile* is the most urgent threat pathogen, and tracking this pathogen's global burden is crucial for disease prevention. This makes the results straightforward and intriguing. However, the following aspects would improve the manuscript:

Sincerely,

Ahmed Abouelkhair

Comments:

L23, 48, 51, 54, 187: *Clostridium difficile* > Clostridium difficile.

L60: emergence of resistance: the evidence is rather sparse. (It is not well documented, like the CDI recurrence.) Some strains show higher MICs, and this doesn't mean resistance.

L69: RT 027> RT027

L69: It should be noticeable in the developed

L71: The author stated the number of CDI infections and deaths in the United States in 2011; however, the most current CDC data stated that the figures decreased to 223,900 cases and 12,800 deaths in 2019. As a result, you should have specified the year for the statistics you provided.

The general writing style may be more straightforward and succinct due to its complexity. A few examples of this are the paragraphs L85–88 in the introduction and L398–401 in the discussion. The discussion denies the newborn's susceptibility to the CDI, despite the introduction's emphasis on it. I would like to see the authors support the babies' resistance to the CDI, as they mentioned in the discussion, in light of what the CDC said.

L150: APC stands for annual percentage change not change percentage.

L272: It should be as well as ASDR and ASMR.

L279: a heavier disease -----the burden, it should be (, and the burden), please rephrase it and the same should be done for L283.

L295: It should be now well described because the relative high incidence, please rephrase it.

L308: It should be (about the impact of CDI in the countries -----), please add the CDI and remove may have.

L327: It should be problem of drug overuse.

L382: *Clostridioides difficile* > Clostridioides difficile

L386: It should be immunoglobulin A (IgA) and G (IgG) antibodies

L397: The sentence (By 3 years of age) should be (While the colonization rate by the age of three).

L416: and the burden of CDI.

September 28, 2024

Dear editors and reviewers,

Thank you for the opportunity to revise our manuscript (Spectrum01290-24R1) entitled "**Global, regional, and national burdens of Clostridium difficile infection from 1990 to 2019: a trend analysis based on the Global Burden of Disease Study 2019**". We appreciate the detailed feedback provided by the reviewers and the editorial team. We have carefully considered all the comments and have made the necessary revisions to address each point raised. Below, we provide a **point-by-point** response to the reviewers' comments and made the relevant revisions in the manuscript.

Reviewer #1 (Comments for the Author):

Excellent work on the revisions, all of my concerns have been addressed! I think this will be a valuable contribution to the field.

Our response:

Thank you for your positive feedback on the revisions. I am glad to hear that all of your concerns have been addressed. I appreciate your support and hope this work will contribute valuable insights to the field.

Reviewer #3 (Comments for the Author):

Clostridium difficile is now called Clostridioides difficile. This has to be corrected throughout the whole text (also the title).

Our response:

Thank you for your valuable feedback. I have corrected all occurrences of "*Clostridium difficile*" to "*Clostridioides difficile*" throughout the text, including the title, as suggested.

Has there been any inclusion or exclusion criteria for included data? Furthermore, I do not understand why the data inclusion ended 2019. Are no newer data available. A lot of different development took place in the meantime e.g. the COVID-19 pandemic and a decline of nosocomial/"hypervirulent" *C. difficile* strains.

Our response:

1. Regarding inclusion or exclusion criteria:

First, I would like to clarify that the GBD database does not directly present data collected from various regions, nor is it a data-upload platform. It gathers data from multiple sources and applies complex modeling techniques to generate estimates of disease burden. The database itself employs extensive standards to ensure data quality and consistency. In our study, we conducted a secondary analysis based on the data modeled by the GBD, and therefore, we did not set any additional inclusion or exclusion criteria.

The GBD collects data from scientific literature, discharge records (from multiple countries), inpatient records, outpatient data, and insurance claims. For *C. difficile*, GBD used the following search string to supplement incidence data on "*C. difficile: clostridium difficile*" AND *diarrhea*[title/abstract] AND (*epidemiolog** OR *incidence* OR *prevalence*) AND (("2017/06/05"[PDat]: "2019/2/7"[PDat])) NOT (*animals*[MeSH] NOT *humans*[MeSH]). They identified 185 studies, of which five met their inclusion criteria. They extracted data points for location, sex, year, and age. The *C. difficile* systematic review flowchart is as follows: [source: PMID:33069326]

We may not be entirely clear about the specific inclusion and exclusion criteria for the data; however, it is worth noting that the GBD study has received widespread recognition and support from international health organizations and experts. Its methodology and data processing have undergone rigorous peer review and evaluation by international experts, ensuring scientific rigor and reliability. Additionally, a flowchart summarizing the input data and methods related to diarrheal diseases in GBD 2019 is shown below [source: PMID:33069326].

We have added a description in the methods section regarding how the GBD database collects and processes data related to *C. difficile*-associated diarrhea. Providing this information will help readers better understand the context and results of our study.

2. Regarding why the data ended in 2019:

The GBD database is updated every two years, and each update involves a comprehensive re-analysis and modeling of several decades of data, rather than simply

adding two years of new data. At the time of conducting this study and submitting the manuscript to this journal, the 2019 data was the latest available, and using 2021 data would have required a complete revision of all results and analyses in the manuscript.

The GBD database has recently been updated to 2021, and we have analyzed this new data. We found that the overall trends remain consistent and do not impact the key conclusions of our study. We analyzed the average changes in DALYs and deaths per 10,000 globally from 2019 to 2021, as well as the average changes in DALYs and deaths per 10,000 among children under 5 and individuals over 70 during the same period. Our findings indicate that the additional data from the past two years had a minimal impact on our results, and this outcome is also consistent with the results of our predictive analysis over the two-year period (**see below figure**). To broaden the applicability of our research data, we have decided to revise the title to "Global, regional, and national burdens of *Clostridioides difficile* infection over recent decades: a trend analysis informed by the Global Burden of Disease Study" This change will help reveal patterns and trends over a longer time frame.

If you believe that using the most recent data is essential, we are willing to make the necessary adjustments and perform a full re-analysis.

Table 1 includes data for certain regions I would emphasize to make an order according to continents rather than alphabetical.

Our response:

Thank you for your suggestion. I have reorganized Table 1 to present the data according to continents rather than in alphabetical order, as recommended. This will help improve the clarity and relevance of the information.

What has also not been discussed is the impact of the before mentioned strains such as ribotype 027 (RT027) and corresponding antimicrobial resistance.

Our response:

Thank you very much for your valuable comments. I have addressed the impact of RT027 in developed regions in the discussion section (L289-L294), specifically focusing on its

role in hospital outbreaks. Regarding antimicrobial resistance, I have revised the manuscript based on the suggestion from Reviewer 4, who pointed out that "some strains show higher MICs, but this doesn't mean resistance." Therefore, I have removed the mention of drug resistance from the introduction to avoid any confusion. I appreciate your insightful feedback, which has been instrumental in improving the manuscript.

Reviewer #4 (Comments for the Author):

L23, 48, 51, 54, 187: *Clostridium difficile* > Clostridium difficile.

Our response:

Thank you for your careful review. I have changed "Clostridium difficile" to italics in all specified locations, as suggested.

L60: emergence of resistance: the evidence is rather sparse. (It is not well documented, like the CDI recurrence.) Some strains show higher MICs, and this doesn't mean resistance.

Our response:

Thank you for your comment. I agree that the evidence regarding resistance is limited and not as well-documented as CDI recurrence. I will revise this statement to better reflect the current understanding.

L69: RT 027> RT027

Our response:

Thank you for your feedback. I have revised the notation to use "RT 027" instead of "RT027" in the specified location.

L69: It should be noticeable in the developed

Our response:

Thank you for your comment. I have made the modifications as suggested in the specified

locations.

L71: The author stated the number of CDI infections and deaths in the United States in 2011; however, the most current CDC data stated that the figures decreased to 223,900 cases and 12,800 deaths in 2019. As a result, you should have specified the year for the statistics you provided.

Our response:

Thank you for your observation. I have specified the year for the statistics provided regarding CDI infections and deaths in the United States. Additionally, I have updated the data to reflect the most current CDC figures from 2019.

The general writing style may be more straightforward and succinct due to its complexity. A few examples of this are the paragraphs L85–88 in the introduction and L398–401 in the discussion. The discussion denies the newborn's susceptibility to the CDI, despite the introduction's emphasis on it. I would like to see the authors support the babies' resistance to the CDI, as they mentioned in the discussion, in light of what the CDC said.

Our response:

Thank you for your valuable comments. We have revised the introduction to emphasize that although infants and newborns are prone to colonization by *Clostridioides difficile*, they rarely exhibit clinical infection, supporting the resistance of infants to CDI.

L150: APC stands for annual percentage change not change percentage.

Our response:

Thank you for your comment. I have corrected the definition of APC to "annual percentage change" as suggested.

L272: It should be as well as ASDR and ASMR.

Our response:

Thank you for pointing that out. I have updated the corresponding sections of the

manuscript to "as well as ASDR and ASMR" as requested.

L279: a heavier disease -----the burden, it should be (, and the burden), please rephrase it and the same should be done for L283.

Our response:

Thank you for your careful review. I have rephrased the sentence as suggested, replacing "a heavier disease --- the burden" with ", and the burden" at both L279 and L283.

L295: It should be now well described because the relative high incidence, please rephrase it.

Our response:

Thank you for your comment. I have rephrased the sentence at L295 to improve clarity and accuracy as suggested.

L308: It should be (about the impact of CDI in the countries -----), please add the CDI and remove may have.

Our response:

Thank you for your suggestion. I have revised the sentence at L308 by adding "CDI" and removing "may have" as recommended.

L327: It should be problem of drug overuse.

Our response:

Thank you for your feedback. I have revised the sentence at L327 to "problem of drug overuse" as suggested.

L382: *Clostridioides difficile* > Clostridioides difficile

Our response:

Thank you for your careful review. I have changed "Clostridium difficile" to italics in specified locations, as suggested.

L386: It should be immunoglobulin A (IgA) and G (IgG) antibodies

Our response:

Thank you for your observation. I have revised the sentence at L386 to correctly state "immunoglobulin A (IgA) and G (IgG) antibodies."

L397: The sentence (By 3 years of age) should be (While the colonization rate by the age of three).

Our response:

Thank you for your suggestion. I have revised the sentence at L397 to "While the colonization rate by the age of three" as recommended.

L416: and the burden of CDI.

Our response:

Thank you for your feedback. I have revised the sentence at L416 to "and the burden of CDI" as suggested.

Thank you again for your effort in our article, and thank you very much for your consideration. We hope that the revision is acceptable and look forward to hearing from you soon. Please contact us for any questions.

With kind regard,

Yours sincerely,

Zhenhua Zhang Prof

Department of Infectious Diseases, Clinical Virus Research Institute, The Second Affiliated Hospital of Anhui Medical University.

Address: Furong Road 678, Hefei 230601, Anhui, China.

Email: zzh1974cn@163.com

Re: Spectrum01290-24R2 (**Global, regional, and national burdens of *Clostridioides difficile* infection over recent decades: a trend analysis informed by the Global Burden of Disease Study**)

Dear Prof. Zhenhua Zhang:

Thank you for the privilege of reviewing your work. Below you will find my comments, instructions from the Spectrum editorial office, and the reviewer comments.

Revision Guidelines

Sincerely,
Yuan Pin Hung
Editor
Microbiology Spectrum

Reviewer #1 (Comments for the Author):

See attached

Reviewer #2 (Comments for the Author):

n/a

Zhou et al. have made modifications and submitted their manuscript for a 3rd round of reviews. While the data appears sound and a valuable contribution to the field, the added discussion points on the correlation between the microbiota as well as immune status and CDI are poorly written, frequently making uncited logical leaps. These missteps, particularly the lack of relevant citations, lead to possibly suspect conclusions. As a reviewer who has now provided feedback on this manuscript for the 3rd time, I am frustrated by the variable advancement in quality, despite the authors' gracious willingness to respond to reviewers. The manuscript also needs a copy editor. For the sake of time, I have not provided feedback on numerous instances where the writing could be streamlined. Below are specific instances where scientific improvement is needed:

Major revisions:

- L81-83 the authors try to link susceptibility of the infant microbiota to CDI with mode of delivery, feeding practices and environmental exposures, but do not appear to offer any citation/evidence to back it up.
- L308 the authors generalize that since CDI is common in Europe, it is therefore more prevalent leading to adverse outcomes in high SDI countries. This is an overgeneralization. Trends in Europe may reflect other countries with high SDI, but there are many countries with high SDI (and CDI incidence) that do not reside in Europe.
- L335 the authors mention lower tier antibiotics. Please provide a short description of what these antibiotics are, similar to what was done with “high tier” earlier in the paper by mentioning specific drugs. I think this is important since the authors at this point do not provide citations to support the lower vs higher tier connection with CDI occurrence.
 - Also, by some standards, metronidazole is a “lower-tier” drug and it is known to have a role in recurrent *C. difficile*.
 - It would seem the more apt distinction is between narrow vs broad spectrum antibiotics rather than a low vs high tier one. However, access to top of the line antibiotics in low SDI countries continues to be a problem. The point is important to make, the text just needs to be revamped.
- Generally, the section on the gut microbiota (L338 and following) is highly reductionistic and doesn't add a lot to the field. I'd characterize it as shockingly sloppy for a microbiota-related paper on *C. difficile* infection...
 - L338-L344 The authors mention several times the “abundance of the gut microbiota”. It would seem they refer not to “abundance”, but rather refer to “diversity” based on context. I do not know of a study that correlates pure abundance of gut microbes with *C. difficile* infectivity. Abundance of particular gut taxa, absolutely yes! But not abundance of all microbes (see also L373). Happy to be proven wrong here though
 - L341 the authors refer to *Prevotella* and *Bacteroides* as “two bacteria” but fail to account for the immense diversity of species/strains in these two genera, and their respective physiological impacts on the host. The Schubert et al. paper cited references OTUs of *Bacteroides/Prevotella*, from what I can tell, not genera in their entirety

- L344 the authors provide a citation for “Functional role of collateral flow in the ischaemic dog heart” in connection to dietary structure of low SDI regions, which is not relevant.
- L344 I am not aware of any studies showing that higher fiber “increases abundance of the microbiota.” Change composition, yes. Change metabolic output, yes. Total abundance, unclear. No citation is provided. And again, authors provide no link to why microbiota abundance would be important, since the vast majority of literature on the topic reference microbiota diversity as a determinant of *C. difficile* colonization.
- L374 The authors loosely try to correlate elderly in hospitals having poor access to probiotics in medical settings with CDI. However, it is unclear if “probiotics” in their general sense have an impact on CDI occurrence. Again, no citation is provided regarding the relevance of this point of conversation. I do not think that publicly available probiotics have a measured, direct impact on CDI. Several companies are developing tailored probiotic formulas specifically for CDI treatment though (some recently approved for use in the U.S.)
- L378-380 “immune system of elderly hosts responds more slowly to microbial molecules that cross the epithelial barrier, leading to rapid development of *C. difficile* toxin-mediated colonic epithelial damage.” The authors’ claim is not backed by a citation, again. And the link between “rapid development” and “slow response” is suspect.

Minor revisions

- L35 and L300 epidemiological change is not specific in what it refers to, thankfully it is defined at L302. But this term is unhelpful because it is a broad category of possible changes, it would be better to just list the causes in L302.
- L56 “widespread” or “abuse”, I don’t think both are needed here
- L62 statement on recurrences needs a citation
- L148, 155, 171 Citations are needed in the methods section for each of these specialized modelling approaches, none are offered.
- L258 the trend being discussed regarding children under 5 years old is not clear, please add an antecedent here so the conclusion being made is understandable
- L304 severe CDI-associated diarrhea -> disease (seems more appropriate in this context. What is high colonic resection? What is the relevance of severe infections to *C. difficile* in connection with antibiotics? Why high dose? Also, what is a high dose of antibiotics?
- L314 abx treatment most important risk factor for CDI. A citation here would be helpful, particularly since it is a well-studied claim
- L385 “fewer specific antibodies” antibodies specific to what? Is there a citation to back up this claim?
- L388 the sentence beginning with “And the incidence...” is not a complete sentence.
- L399-following. The authors provide a long list of conditions that are risk factors for children, is this needed? Maybe choose a couple that are relevant to the findings of this manuscript?
- L403 “Children” should be lower case
- L408 “manage this population” -> “manage infections in this population”

- L415 and following. The authors discuss use of DALYs and its drawbacks. This reads like a limitation to me and should be moved after L425

Oct 6th, 2024

Dear Dr. Yuan Pin Hung,

Thank you for the opportunity to re-review the manuscript titled “**Global, regional, and national burdens of *Clostridioides difficile* infection over recent decades: a trend analysis informed by the Global Burden of Disease Study**” (ID: Spectrum01290-24R2) by Jianmei Zhou, Jie Zhu, Pengyue Zhang, Chunhui Tao, Xiaodan Hong, Zhenhua Zhang

I have carefully reviewed the revised version and am pleased to see that the authors have addressed all of my previous comments satisfactorily.

I believe the manuscript has been significantly improved and is now suitable for publication in the microbiology spectrum. I do not have any further suggestions at this time.

Please let me know if there is anything else you would like me to consider.

Sincerely,

Ahmed Abouelkhair

November 30, 2024

Dear editors and reviewers,

Thank you for your continued time and effort in reviewing our manuscript (Spectrum01290-24R2) entitled "**Global, regional, and national burdens of *Clostridioides difficile* infection over recent decades: a trend analysis informed by the Global Burden of Disease Study**". We greatly appreciate your constructive feedback, which has been invaluable in improving the quality of our work. We have carefully reviewed your suggestions and have made the necessary revisions to ensure clearer writing and stronger scientific support for the points made in the manuscript. Below, we provide a point-by-point response to the reviewers' comments and made the relevant revisions in the manuscript.

Reviewer #1 (Comments for the Author):

Major revisions:

1. L81-83 the authors try to link susceptibility of the infant microbiota to CDI with mode of delivery, feeding practices and environmental exposures, but do not appear to offer any citation/evidence to back it up.

Our response:

Thank you for your valuable comments. We apologize for not providing sufficient references in the relevant section. Regarding the susceptibility of the infant microbiota to CDI, existing studies have shown that the high colonization rate of *C. difficile* in infants is closely related to their gut microbiota. The early microbiota development in infants is influenced by various factors, including delivery method, diet, and various maternal factors. These points are discussed in the review article "*Clostridioides difficile* and the Microbiota Early in Life" (PMID: 34791400, Figure 1), which I have now cited in the main text. Thank you again for your feedback, and we will revise the text accordingly.

2. L308 the authors generalize that since CDI is common in Europe, it is therefore more prevalent leading to adverse outcomes in high SDI countries. This is an

overgeneralization. Trends in Europe may reflect other countries with high SDI, but there are many countries with high SDI (and CDI incidence) that do not reside in Europe.

Our response:

Thank you for your valuable comments. We apologize for not clarifying this point sufficiently. The conclusion of our study is that regions with higher SDI are generally associated with a heavier burden of CDI infections. Therefore, we included Europe as an example to support this conclusion. However, I fully agree with your point that relying solely on Europe as an example may be an overgeneralization. To address this, A review and meta-analysis on CDI includes CDI incidence data associated to healthcare facility from 41 countries worldwide, and through comparison, it was found that CDI incidence is higher in high-income North America (**PMID: 30603078**). This addition further supports our conclusion.

3. L335 the authors mention lower tier antibiotics. Please provide a short description of what these antibiotics are, similar to what was done with “high tier” earlier in the paper by mentioning specific drugs. I think this is important since the authors at this point do not provide citations to support the lower vs higher tier connection with CDI occurrence.

- Also, by some standards, metronidazole is a “lower-tier” drug and it is known to have a role in recurrent *C. difficile*.
- It would seem the more apt distinction is between narrow vs broad spectrum antibiotics rather than a low vs high tier one. However, access to top of the line antibiotics in low SDI countries continues to be a problem. The point is important to make, the text just needs to be revamped.

Our response:

Thank you for your valuable suggestion. Based on your feedback, I reviewed the recent article "Antibiotics and healthcare facility-associated *Clostridioides difficile* infection: systematic review and meta-analysis 2020 update"(**PMID: 33787887**). This study indicates that clindamycin, carbapenems, and third- and fourth-generation

cephalosporins are most strongly associated with healthcare facility-associated *Clostridioides difficile* infection, while antibiotics such as tetracycline, sulfonamides, and macrolides show a weaker association with CDI. In light of this, I have revised the text to distinguish antibiotics as narrow spectrum versus broad spectrum, rather than low tier versus high tier, as suggested. I have also provided a detailed list of antibiotics that are available in low SDI regions. Relevant citations have been added to the manuscript.

4. Generally, the section on the gut microbiota (L338 and following) is highly reductionistic and doesn't add a lot to the field. I'd characterize it as shockingly sloppy for a microbiotarelated paper on *C. difficile* infection...

- L338-L344 The authors mention several times the “abundance of the gut microbiota”. It would seem they refer not to “abundance”, but rather refer to “diversity” based on context. I do not know of a study that correlates pure abundance of gut microbes with *C. difficile* infectivity. Abundance of particular gut taxa, absolutely yes! But not abundance of all microbes (see also L373). Happy to be proven wrong here though

Our response:

Thank you for your valuable feedback. You are absolutely correct in pointing out the distinction between "abundance" and "diversity." I did indeed use the term inaccurately, and "diversity" is the more appropriate term to describe the relevant characteristics of the gut microbiota. I have made the necessary revisions in the manuscript based on your suggestion.

- L341 the authors refer to *Prevotella* and *Bacteroides* as “two bacteria” but fail to account for the immense diversity of species/strains in these two genera, and their respective physiological impacts on the host. The Schubert et al. paper cited references OTUs of *Bacteroides/Prevotella*, from what I can tell, not genera in their entirety

Our response:

Thank you for your valuable feedback. You are correct that referring to *Prevotella* and *Bacteroides* as “two bacteria” was inappropriate. In discussing the potential reasons for the lower CDI burden in low SDI regions, our intention was to approach this issue from the perspective of the gut microbiota in these populations. To support this, we referenced the study by Schubert et al. to highlight *Bacteroides* as a protective gut microbiota associated with CDI. However, the original paper did not specifically address the relationship between OTUs of *Bacteroides* and CDI. Moreover, most studies describe the gut microbiota at the genus level, without delving into the specific species or strains within these genera. Additionally, we acknowledge that the logic and clarity of this section in the manuscript may not have been fully clear, and I have made the necessary revisions to improve this, to help the readers better understand the context.

- L344 the authors provide a citation for “Functional role of collateral flow in the ischaemic dog heart” in connection to dietary structure of low SDI regions, which is not relevant.

Our response:

Thank you for your valuable feedback. I apologize for the incorrect citation in L344. I intended to reference the study by De Filippo et al. (PMID: 20679230), which discusses how higher fiber diets in low SDI regions, such as Africa, lead to differences in gut microbiota composition compared to high-SDI regions. I have corrected this citation in the manuscript to accurately reflect the intended reference.

- L344 I am not aware of any studies showing that higher fiber “increases abundance of the microbiota.” Change composition, yes. Change metabolic output, yes. Total abundance, unclear. No citation is provided. And again, authors provide no link to why microbiota abundance would be important, since the vast majority of literature on the topic reference microbiota diversity as a determinant of *C. difficile* colonization.

Our response:

Thank you for your valuable comment. The use of the term "abundance" was indeed inappropriate. What I intended to convey was that a higher-fiber diet promotes the growth of protective gut microbiota species and increases the diversity of the gut microbiome, which in turn enhances the gut's resistance to CDI. I have made the necessary corrections in the manuscript and added relevant references to support this point.

- L374 The authors loosely try to correlate elderly in hospitals having poor access to probiotics in medical settings with CDI. However, it is unclear if "probiotics" in their general sense have an impact on CDI occurrence. Again, no citation is provided regarding the relevance of this point of conversation. I do not think that publicly available probiotics have a measured, direct impact on CDI. Several companies are developing tailored probiotic formulas specifically for CDI treatment though (some recently approved for use in the U.S.)

Our response:

Thank you for your valuable comments. I apologize for any misunderstanding caused by my wording. In line 374, the original sentence "Elderly individuals generally have lower overall gut microbiota abundance and lower abundance of probiotics in medical settings" was intended to convey that elderly individuals generally have lower abundance of protective gut microbiota species in their gut microbiota in medical settings (**PMID: 20571116**), rather than referring to difficulties in accessing probiotics. I appreciate the reviewer's point regarding the role of probiotics in CDI, but this was not the focus of my statement. To clarify, I have revised the manuscript to more accurately express this idea and have added relevant citations for support.

- L378-380 "immune system of elderly hosts responds more slowly to microbial molecules that cross the epithelial barrier, leading to rapid development of C. difficile toxin-mediated colonic epithelial damage." The authors' claim is not backed by a citation, again. And the link between "rapid development" and "slow response" is suspect.

Our response:

Thank you for pointing this out, and I apologize for any confusion caused by the original statement. My intention was to highlight that the increased susceptibility of elderly individuals to CDI may, in part, be due to impaired intestinal barrier function (PMID: 28956703; PMID: 31369869). I recognize that the original phrasing may have been unclear. I have revised the sentence to clarify this point and included relevant citations to support the statement.

Minor revisions

1. L35 and L300 epidemiological change is not specific in what it refers to, thankfully it is defined at L302. But this term is unhelpful because it is a broad category of possible changes, it would be better to just list the causes in L302.

Our response:

Thank you for the valuable comments. To further clarify, I would like to explain that "epidemiological change" is a standard term widely used in the field of epidemiology. Specifically, the main drivers behind changes in disease burden are threefold: age structure, population growth, and epidemiological changes. Each factor affects disease burden differently across regions and time periods. The purpose of the decomposition analysis model is to determine which factor is the primary driver of changes in disease burden. Our study indicates that in high SDI regions, the changes in CDI burden are mainly driven by epidemiological changes. These epidemiological changes may be due to the prevalence of highly virulent strains, antibiotic exposure in high SDI regions.

2. L56 "widespread" or "abuse", I don't think both are needed here.

Our response:

Thank you for your insightful comment. I agree that using both "widespread" and "abuse" may be redundant. I have revised the sentence to use only one of the terms for clarity and conciseness.

3. L62 statement on recurrences needs a citation.

Our response:

Thank you for your valuable comment. I agree that the statement on recurrences requires proper citation. I have added the relevant reference to support this point in the manuscript.

4. L148, 155, 171 Citations are needed in the methods section for each of these specialized modelling approaches, none are offered.

Our response:

Thank you for your valuable feedback. I agree with your suggestion that citations are needed for the specialized modeling approaches mentioned in the methods section. I have added the appropriate references for each of these methods in the revised manuscript.

5. L258 the trend being discussed regarding children under 5 years old is not clear, please add an antecedent here so the conclusion being made is understandable

Our response:

Thank you for your valuable feedback. I agree that the trend regarding children under 5 years old needs clearer contextualization. I have added the necessary antecedents to ensure the conclusion is more understandable.

6. L304 severe CDI-associated diarrhea -> disease (seems more appropriate in this context. What is high colonic resection? What is the relevance of severe infections to *C. difficile* in connection with antibiotics? Why high dose? Also, what is a high dose of antibiotics?

Our response:

Thank you for your valuable feedback on our manuscript. In this section, our initial intention was to highlight that the heavier burden of *Clostridioides difficile* in high SDI regions might be attributed to the use of broad-spectrum antibiotics. However, we

realized that the reference cited in this context was not sufficiently accurate, leading to a lack of clarity in the statement. We have revised this part accordingly and included more appropriate references to support this argument.

7. L314 abx treatment most important risk factor for CDI. A citation here would be helpful, particularly since it is a well-studied claim

Our response:

Thank you for your valuable feedback. We agree that a citation is needed to support the statement regarding antibiotic treatment being the most important risk factor for CDI. We have added the relevant citation in the manuscript to strengthen this claim.

8. L385 “fewer specific antibodies” antibodies specific to what? Is there a citation to back up this claim?

Our response:

Thank you for your comment, and I apologize for the unclear phrasing. The phrase "fewer specific antibodies" refers to IgG antibodies against *C. difficile* toxin A. Studies have shown that hospitalized patients with high levels of anti-toxin A IgG do not develop diarrhea after CDI (PMID: 10666429). Additionally, serum IgG and IgM responses to toxin A are crucial for protecting patients from CDI and preventing recurrence (PMID: 11213096). Other studies have indicated that, compared to younger individuals, healthy elderly subjects have lower serum IgG levels against *C. difficile* toxin (PMID: 7253967, PMID: 7924215). I have added the relevant citations in the revised manuscript to support this statement.

9. L388 the sentence beginning with “And the incidence...” is not a complete sentence.

Our response:

Thank you for pointing that out. I have revised the sentence beginning with "And the incidence..." to ensure it is a complete sentence in the manuscript.

10. L399-following. The authors provide a long list of conditions that are risk factors for children, is this needed? Maybe choose a couple that are relevant to the findings of this manuscript?

Our response:

Thank you for your valuable feedback. I will revise this section as per your suggestion, selecting only a few risk factors that are relevant to the findings of this manuscript.

11. L403 "Children" should be lower case

Our response:

Thank you for your valuable comment. I agree with your suggestion, and I have corrected "Children" to lowercase in the revised manuscript.

12. L408 "manage this population" -> "manage infections in this population"

Our response:

Thank you for the suggestion. I have revised the phrase "manage this population" to "manage infections in this population" in the manuscript. The necessary changes have been made accordingly.

13. L415 and following. The authors discuss use of DALYs and its drawbacks. This reads like a limitation to me and should be moved after L425

Our response:

Thank you for your valuable feedback. I agree that the discussion on DALYs and its drawbacks should be considered as part of the study's limitations and placed later in the manuscript. I have moved this section to after L425 to improve the flow of the text.

Reviewer #4 (Comments for the Author):

I have carefully reviewed the revised version and am pleased to see that the authors have addressed all of my previous comments satisfactorily.

I believe the manuscript has been significantly improved and is now suitable for

publication in the microbiology spectrum. I do not have any further suggestions at this time. Please let me know if there is anything else you would like me to consider.

Our response:

Thank you for your thorough review of our revised manuscript and for your positive feedback. We are pleased to hear that you believe the manuscript has been significantly improved and is now suitable for publication in Microbiology Spectrum. We greatly appreciate your support and encouragement.

Thank you again for your effort in our article, and thank you very much for your consideration. We hope that the revision is acceptable and look forward to hearing from you soon. Please contact us for any questions.

With kind regard,

Yours sincerely,

Zhenhua Zhang Prof

Department of Infectious Diseases, Clinical Virus Research Institute, The Second Affiliated Hospital of Anhui Medical University.

Address: Furong Road 678, Hefei 230601, Anhui, China.

Email: zzh1974cn@163.com

Re: Spectrum01290-24R3 (**Global, regional, and national burdens of *Clostridioides difficile* infection over recent decades: a trend analysis informed by the Global Burden of Disease Study**)

Dear Prof. Zhenhua Zhang:

Thank you for the privilege of reviewing your work. Below you will find my comments, instructions from the Spectrum editorial office, and the reviewer comments.

Revision Guidelines

Sincerely,
Yuan Pin Hung
Editor
Microbiology Spectrum

Reviewer #1 (Comments for the Author):

Please see the attached word document

Zhou et al. have made modifications and submitted their manuscript for a 4th round of reviews. The data/analysis appears to be a sound/valuable contribution to the field, and the discussion much improved. For the sake of time, I have limited my responses to those their response to previous my critiques. I have not provided feedback on instances where the writing could be further streamlined, except when there is scientific concern. While the discussion of the microbiota and CDI is much improved, there is one remaining concern about this section as well as a response to the discussion on epidemiologic change.

1. Microbiota and CDI

- a. L317 “Bacteroides and Prevotella [sic] are considered protective gut microbiota species...”
 - i. In response to my previous statement that this section of the discussion is overly generalized, the others state “the original paper [Schubert et al.] does not address the relationship between OTUs and CDI.” I kindly suggest they review Figures 2 and 3 of the paper cited (Schubert et al. <https://doi.org/10.1128/mbio.01021-14>) for an analysis of the relationship between OTUs and CDI-specific diarrhea. Schubert et al. are careful to distinguish between the genera *Bacteroides* and *Prevotella* and particular OTUs within those taxa. It is well known in the microbiota field that short read sequencing has difficulty identifying particular species, so operational taxonomic units (OTUs) are considered loosely, even if at times arbitrarily, equivalent to species given the vast variety of taxa present within a particular genus.
 - ii. The authors could correct this by writing “Particular species/OTUs of the *Bacteroides* and *Prevotella* genera are considered protective”
- b. L318 “Firmicutes are considered harmful species”
 - i. I kindly suggest that the authors review Figures 2 and 3 of the paper cited (Schubert et al. <https://doi.org/10.1128/mbio.01021-14>) for an analysis showing numerous Firmicutes that are anticorrelated with CDI diarrhea (see Lachnospiraceae, Ruminococcus and Odoribacter OTUs). Please also see https://scholar.google.com/scholar?hl=en&as_sdt=0%2C23&q=firmicutes+AND+cdi+AND+protective&btnG= for numerous other references that show the protective role of members of the Firmicutes phylum (e.g., *Clostridium scindens*). It is odd given the diversity of the gut microbiota that an entire bacterial phylum be designated as harmful in the context of CDI.
 - ii. The authors could correct this by being specific, such as by writing some of the Firmicutes species/OTUs listed in the Schubert paper that are correlated with CDI resistance/susceptibility
- c. If the authors wish to discuss the Firmicutes phylum, I kindly suggest they conform to the new nomenclature guidelines for bacterial phyla <https://ncbiinsights.ncbi.nlm.nih.gov/2021/12/10/ncbi-taxonomy-prokaryote-phyla-added/> For example, Firmicutes is now termed Bacillota.
- d. Regardless of their discussion of particular bacterial taxa, there is a number of articles discussing the impact of dietary fiber on CDI that may offer more direct support for their conclusions (see

https://scholar.google.com/scholar?hl=en&as_sdt=0%2C23&q=dietary+fiber+AND+cdi&btnG=). A particular example of note is <https://doi.org/10.1038/s41564-018-0150-6>

2. Epidemiological change

- a. This reviewer sincerely appreciates the authors engagement in conversation regarding use of “epidemiological change.” After diving further into the literature, I learned that it is a commonly used term in the field of epidemiology. But given that it took me over an hour to understand the definition even as an expert in the gut microbiota/infectious disease, I do worry about whether using it is informative or just academic jargon. However, since the authors define it at least once in the document and still want to use the term, I think it is okay as is.

December 18, 2024

Dear editors and reviewers,

Thank you for your continued time and effort in reviewing our manuscript (Spectrum01290-24R3) entitled "**Global, regional, and national burdens of *Clostridioides difficile* infection over recent decades: a trend analysis informed by the Global Burden of Disease Study**". We greatly appreciate your constructive feedback, which has been invaluable in improving the quality of our work. We have carefully reviewed your suggestions and have made the necessary revisions. In addition, we have streamlined the manuscript to further improve the writing. Below, we provide a point-by-point response to the reviewers' comments and made the relevant revisions in the manuscript.

Reviewer #1 (Comments for the Author):

1. Microbiota and CDI

a. L317 "Bacteroides and Prevotella [sic] are considered protective gut microbiota species..."

i. In response to my previous statement that this section of the discussion is overly generalized, the others state "the original paper [Schubert et al.] does not address the relationship between OTUs and CDI." I kindly suggest they review Figures 2 and 3 of the paper cited (Schubert et al. <https://doi.org/10.1128/mbio.01021-14>) for an analysis of the relationship between OTUs and CDI-specific diarrhea. Schubert et al. are careful to distinguish between the genera Bacteroides and Prevotella and particular OTUs within those taxa. It is well known in the microbiota field that short read sequencing has difficulty identifying particular species, so operational taxonomic units (OTUs) are considered loosely, even if at times arbitrarily, equivalent to species given the vast variety of taxa present within a particular genus.

ii. The authors could correct this by writing "Particular species/OTUs of the Bacteroides and Prevotella genera are considered protective"

Our response:

Thank you for your suggestion. We have carefully reviewed Figures 2 and 3 of the Schubert et al. paper, which indeed distinguish between the genera *Bacteroides* and *Prevotella* and particular OTUs within those taxa. In response to your recommendation, we will revise the manuscript to state that "particular species/OTUs of the *Bacteroides* and *Prevotella* genera are considered protective."

b. L318 "Firmicutes are considered harmful species"

- i. I kindly suggest that the authors review Figures 2 and 3 of the paper cited (Schubert et al. <https://doi.org/10.1128/mbio.01021-14>) for an analysis showing numerous Firmicutes that are anticorrelated with CDI diarrhea (see Lachnospiraceae, *Ruminococcus* and *Odoribacter* OTUs). Please also see

https://scholar.google.com/scholar?hl=en&as_sdt=0%2C23&q=firmicutes+AND+cdi+AND+protective&btnG= for numerous other references that show the protective role of members of the Firmicutes phylum (e.g., *Clostridium scindens*). It is odd given the diversity of the gut microbiota that an entire bacterial phylum be designated as harmful in the context of CDI.

- ii. The authors could correct this by being specific, such as by writing some of the Firmicutes species/OTUs listed in the Schubert paper that are correlated with CDI resistance/susceptibility.

Our response:

Thank you for your valuable suggestion. Given the diversity of the gut microbiota, it is inappropriate to designate the entire Firmicutes phylum as harmful in the context of CDI. We have reviewed the relevant figures in the Schubert et al. Paper, and based on their study, we have listed specific Firmicutes species/OTUs associated with CDI susceptibility, such as *Clostridium XI* (OTU39), *Erysipelotrichaceae* (OTU22), and *Streptococcus* (OTU10). This will allow for a more accurate representation of the

relationship between Firmicutes and CDI.

- c. If the authors wish to discuss the Firmicutes phylum, I kindly suggest they conform to the new nomenclature guidelines for bacterial phyla <https://ncbiinsights.ncbi.nlm.nih.gov/2021/12/10/ncbi-taxonomy-prokaryote-phylaadded/> For example, Firmicutes is now termed Bacillota.

Our response:

Thank you for your suggestion. We will update the manuscript to conform to the new nomenclature guidelines and refer to the Firmicutes phylum as Bacillota, as recommended.

- d. Regardless of their discussion of particular bacterial taxa, there is a number of articles discussing the impact of dietary fiber on CDI that may offer more direct support for their conclusions (see

[https://scholar.google.com/scholar?hl=en&as_sdt=0%2C23&q=dietary+fiber+AND+cdi&btnG=\)](https://scholar.google.com/scholar?hl=en&as_sdt=0%2C23&q=dietary+fiber+AND+cdi&btnG=)). A particular example of note is <https://doi.org/10.1038/s41564-018-0150-6>

Our response:

Thank you for providing the relevant literature on the impact of dietary fiber on CDI, such as **PMID: 29686297**. In response to your suggestion, we have included the relevant references in the manuscript to further support our conclusions. We appreciate your valuable input.

2. Epidemiological change

- a. This reviewer sincerely appreciates the authors engagement in conversation regarding use of “epidemiological change.” After diving further into the literature, I learned that it is a commonly used term in the field of epidemiology. But given that it took me over an hour to understand the definition even as an expert in the gut microbiota/infectious disease, I do worry about whether using it is informative or just academic jargon. However, since the authors define it at least once in the document and still want to use

the term, I think it is okay as is.

Our response:

Thank you for your thoughtful feedback. We appreciate your time and effort in exploring the term "epidemiological change." We understand your concern about its clarity and have ensured that it is clearly defined in the manuscript. Since you find the definition satisfactory, we will retain the term as is. Thank you again for your valuable input.

Thank you again for your effort in our article, and thank you very much for your consideration. We hope that the revision is acceptable and look forward to hearing from you soon. Please contact us for any questions.

With kind regard,

Yours sincerely,

Zhenhua Zhang Prof

Department of Infectious Diseases, Clinical Virus Research Institute, The Second
Affiliated Hospital of Anhui Medical University.

Address: Furong Road 678, Hefei 230601, Anhui, China.

Email: zzh1974cn@163.com

Re: Spectrum01290-24R4 (**Global, regional, and national burdens of *Clostridioides difficile* infection over recent decades: a trend analysis informed by the Global Burden of Disease Study**)

Dear Prof. Zhenhua Zhang:

Thank you for the privilege of reviewing your work. Below you will find my comments, instructions from the Spectrum editorial office, and the reviewer comments.

Revision Guidelines

Sincerely,
Yuan Pin Hung
Editor
Microbiology Spectrum

Reviewer #1 (Comments for the Author):

I have no further critiques for the paper, your responsiveness has been appreciated!

Reviewer #4 (Comments for the Author):

Minor changes:

L46 and 347: *Clostridium difficile* (ITALIC)> Clostridium difficile.

L57: Please only use one since reinfection is the same as recurrence.

L64: developed what? Could you please elaborate?

L74: Please rephrase this sentence for clarity, as it repeats L75.

L88: Methods> method

L129-131: This sentence has to be rephrased because it is repetitious.

L320: some>some!

L331-333: Please rephrase this section for clarity, as it repeats L341-343.

L356: Since this is the first time adding, please explain what IBD stands for.

L378: data by age group. And the, kindly fix it.

December 29, 2024

Dear editors and reviewers,

Thank you for your continued time and effort in reviewing our manuscript (Spectrum01290-24R4) entitled "**Global, regional, and national burdens of *Clostridioides difficile* infection over recent decades: a trend analysis informed by the Global Burden of Disease Study**". We greatly appreciate your constructive feedback, which has been invaluable in improving the quality of our work. We have carefully reviewed your suggestions and have made the necessary revisions. Below, we provide a point-by-point response to the reviewers' comments and made the relevant revisions in the manuscript.

Reviewer #4 (Comments for the Author):

Minor changes:

1. L46 and 347: Clostridium difficile (ITALIC)> Clostridium difficile.

Our response:

Thank you for your valuable feedback. I have made the necessary corrections to ensure that *Clostridium difficile* is italicized throughout the manuscript as per your suggestion.

2. L57: Please only use one since reinfection is the same as recurrence.

Our response:

Thank you for your insightful comment. I have revised the manuscript to use only one term, as "reinfection" and "recurrence" are indeed synonymous in this context.

3. L64: developed what? Could you please elaborate?

Our response:

Thank you for your suggestion. I have revised the sentence to clarify that the trend is particularly noticeable in the developed countries.

4. L74: Please rephrase this sentence for clarity, as it repeats L75.

Our response:

Thank you for your feedback. I have revised the sentence for clarity and removed the repetition with L75.

5. L88: Methods> method.

Our response:

Thank you for your comment. I have corrected "Method" to "Methods" as per your suggestion.

6. L129-131: This sentence has to be rephrased because it is repetitious.

Our response:

Thank you for your suggestion. I have revised the sentence to eliminate the repetition and clarify the explanation.

7. L320: some>some1.

Our response:

Thank you for your comment. I have corrected "some1" to "some" as per your suggestion.

8. L331-333: Please rephrase this section for clarity, as it repeats L341-343.

Our response:

Thank you for your feedback. I have rephrased this section to improve clarity and avoid repetition with lines 341-343.

9. L356: Since this is the first time adding, please explain what IBD stands for.

Our response:

Thank you for your comment. I have added the full form of IBD as "inflammatory bowel disease" upon its first mention in the manuscript.

10. L378: data by age group. And the, kindly fix it.

Our response:

Thank you for your feedback. I have revised the sentence to improve the flow.

Thank you again for your effort in our article, and thank you very much for your consideration. We hope that the revision is acceptable and look forward to hearing from you soon. Please contact us for any questions.

With kind regard,

Yours sincerely,

Zhenhua Zhang Prof

Department of Infectious Diseases, Clinical Virus Research Institute, The Second
Affiliated Hospital of Anhui Medical University.

Address: Furong Road 678, Hefei 230601, Anhui, China.

Email: zzh1974cn@163.com

Re: Spectrum01290-24R5 (**Global, regional, and national burdens of *Clostridioides difficile* infection over recent decades: a trend analysis informed by the Global Burden of Disease Study**)

Dear Prof. Zhenhua Zhang:

Your manuscript has been accepted, and I am forwarding it to the ASM production staff for publication. Your paper will first be checked to make sure all elements meet the technical requirements. ASM staff will contact you if anything needs to be revised before copyediting and production can begin. Otherwise, you will be notified when your proofs are ready to be viewed.

Sincerely,
Yuan Pin Hung
Editor
Microbiology Spectrum

Reviewer #3 (Comments for the Author):

Dear authors,

since my reviewer comments were addressed properly, I do not have any more questions left.

With the best regards

Reviewer #4 (Comments for the Author):

I have no more comments for the manuscript; I appreciate your response!